



# 1 Geomorphic influences on the contribution of vegetation to soil
# 2 C accumulation and accretion in *Spartina alterniflora* marshes

Tracy Elsey-Quirk[1], Viktoria Unger[2]
[1]Department of Oceanography and Coastal Sciences, Louisiana State University, Baton Rouge, LA 70803, USA
[2]University of Rostock, Rostock, 18051, Germany
*Correspondence to*: Tracy Quirk (tquirk@lsu.edu)
**Abstract.** Environmental conditions have a strong influence on rates plant productivity and decomposition. In salt
marshes, hydrology and salinity are important regulators of plant and soil processes, which, in turn, can influence the
rate at which marsh ecosystems accumulate C and adjust to sea-level rise. For this study, we examined the influence
of multivariate environmental conditions on belowground ingrowth (roots + rhizomes), decomposition and biomass
in marshes dominated by *Spartina alterniflora* across two estuaries and a range of geomorphic settings. Secondly, we
examined the influence of belowground plant biomass to soil C density, and C (labile and refractory) accumulation
and accretion rates. Study locations occupied a full range of tidal elevations from below mean low water to above
mean high water. Salinities ranged from 7 – 40, and soil properties also varied across marshes. While many of the
environmental parameters were correlated across marshes, belowground ingrowth of *S. alterniflora* was negatively
influenced by mean low water height, such that root growth increased with more drainage. Belowground decay rate
increased with increasing salinity, but ultimately the percent of mass remaining was similar across marshes, averaging
59 ± 1%. Above- and belowground biomass dynamics were estuary-dependent. In the coastal lagoon estuary, less
flooding and a higher sedimentation rate favored above-and belowground biomass, which, in turn, increased soil C
accumulation and accretion rates. Biomass dynamics in the coastal plain estuary, for the most part, were unrelated to
environmental predictor variables, and had little influence on the accumulation of soil C or accretion rate. These
findings indicate that mineral sedimentation is of utmost importance for promoting belowground biomass and soil C
accumulation in sediment-limited systems while in minerogenic systems, belowground biomass may not scale with C
accumulation and accretion, which may be influenced more by smaller submillimetre-sized C particles.

## 26 1 Introduction

Salt marshes are among the most productive ecosystems on Earth where over half of the annual plant biomass
produced can occur belowground in the form of rhizomes and roots (Valiela et al., 1976; Gallagher and Plumley 1979;
Schubauer and Hopkinson 1984). Belowground biomass of perennial marsh grasses, sedges and rushes is considered
a major contributor to soil organic matter and organic carbon (C) as it accumulates and undergoes relatively slow
decomposition. The preservation of organic matter in marsh soil is promoted by both anaerobic conditions and
refractory C compounds of plant tissues, primarily associated with cell walls (Valiela et al., 1979; Wilson et al., 1986).
However, the same anaerobic conditions that can slow decomposition may constrain the production of roots and
rhizomes with low oxygen and high sulfide concentrations. Hydrology, therefore, is predicted to be of primary
importance influencing root growth and belowground biomass. Root growth of *Spartina alterniflora*, a common low
marsh species along much of the Atlantic and Gulf coasts of the U.S., into new substrate (i.e., ingrowth) has been
shown to be lower in marsh areas with lower redox potential and higher soil moisture (Blum 1993). Interestingly,
these same low redox conditions that limit root ingrowth, can promote high belowground biomass (Gallagher and
Plumley 1979; Dame and Kenny 1986 and references therein). This is hypothesized to be due to greater investment
in belowground production (Hopkinson and Schubauer 1984) and less photosynthate transfer from underground
rhizomes throughout the year, which results in lower aboveground biomass and higher belowground biomass in short-
form interior populations of *S. alterniflora* than tall-form populations growing along creekbank (Gallagher and Kibby



1981). While hydrology can play an important role in biomass production, organic matter decay rates have been
shown to be more variable and less related to flooding dynamics (Blum 1993).

Ultimately, the balance of plant productivity and organic matter preservation contributes to peat formation and soil C
accumulation, which is also highly dependent on mineral sedimentation, both for promoting conditions that enhance
plant productivity relative to decay and contributing directly to the burial and preservation of organic matter and
accretion (DeLaune et al., 1990a; Mendelssohn and Kuhn 2003; Elsey-Quirk et al., 2011; Kirwan and Megonigal
2013; Boyd et al., 2017). Positive relationships between organic matter accumulation, mineral sedimentation, and
accretion are well established with organic matter contributing disproportionately more to soil volume (reviewed in
Turner et al., 2000; Neubauer 2008). However, there have been few empirical studies on the direct contribution of
plant productivity and biomass to soil C and accretion in natural marshes. A greenhouse experiment illustrated that
flooding stimulated surface root growth of a high marsh grass, *Spartina patens*, which was proposed to be a mechanism
by which flooding enhanced organic matter accumulation and accretion (Nyman et al., 2006). In mangrove forests,
the contribution of root production to vertical soil expansion was highly site-specific, with a stronger positive
relationship in peat soils as compared to organic and mineral soil types (Cahoon et al., 2006). These studies suggest
that environmental conditions and soil organic matter content play important roles in the relationship between plant
roots and accretion. In salt marshes, localized environmental conditions such as hydrology, salinity, sediment
availability and soil properties are predicted to play a key role in influencing relationships between biotic processes
and rates of C accumulation and accretion. Determining direct relationships between abiotic factors and vegetation
response across natural systems, however, is challenged by inherent correlations and collinearities. Yet, understanding
relationships across natural systems is extremely important as multiple abiotic conditions will determine the ability of
marshes to accumulate and store C and adjust their elevation relative to sea-level.

While all marshes were formed in low-energy coastal environments, they occur in a wide range geomorphic settings
ranging from back barrier to deltaic estuaries. Different landscape and coastal processes have influenced their origin
and development (Orson et al., 1985) and environmental factors such as tidal range, sediment supply, salinity, and
nutrient loads may all differ both within and among these estuarine types. It is these interacting geomorphological
characteristics that can affect marsh vulnerability to sea level rise and/or erosion. For example, microtidal marshes in
coastal lagoons are considered vulnerable to submergence with accelerating sea-level rise primarily due to limited
sediment supply and tidal hydrology that may be too weak or infrequent for sufficient sediment transport and
deposition (Reed et al., 2008; Kirwan & Guntenspergen 2010; Ganju et al., 2017). Accretion in these systems depend
primarily on organic matter produced by marsh macrophytes, though our understanding of how geomorphology
influences plant and decay processes and their contribution to biomass structure and longer-term C accumulation and
accretion rates is relatively limited, and largely based on models. Accretion and soil C accumulation can be influenced
by sediment and plant litter deposition, erosion, compaction, as well as root growth and decomposition. Short-term
marsh surface accretion rates ($\leq$ 5 yrs) have been shown to be strongly related to allochthonous sediment inputs rather
than autochthonous plant production (Cahoon et al., 1994; 1996; Saintilan et al., 2013), while C from root biomass



may be more important at depth and over time (Saintilan et al., 2013). Given the potential importance of belowground
plant and organic matter dynamics to soil structure in the marsh interior where sedimentation may be lower, and where
conversion to open water due to sea-level rise may be most prevalent (e.g., DeLaune et al., 1994), identifying
environmental constraints on growth and decay and relationships to C accumulation and accretion will be important
for informing marsh management and restoration.

Though geomorphic influences on the relationship between biotic processes and soil C and accretion are not well
constrained in natural marshes, geomorphic differences in sediment availability has been shown to have a large
influence on organic matter accumulation and accretion (Ibañez et al. 2010). A study of subsiding marshes of the
deltaic region of Louisiana, U.S. illustrated that organic matter accumulation was the dominant factor controlling
accretion, both of which were hypothesized to be constrained by flooding stress and/or low mineral sediment
availability on root production (Nyman et al. 1993). In a previous study, we found geomorphic differences in mineral
sediment accumulation rates influenced higher organic matter accumulation and accretion rates in a coastal plain
estuary as compared to a coastal lagoon (Boyd et al., 2017). A strong relationship between mineral sedimentation and
organic accumulation implies that plant productivity and biomass may be greater in marshes with greater sediment
availability, as sediment deposition creates favorable conditions for plant growth (Mendelssohn and Kuhn 2003).
Further, we found that sediment accumulation rate was positively related to rates of labile C, but not refractory C
accumulation (Unger et al., 2016). Geomorphology, therefore, influenced the relative chemical recalcitrance of soil
C, with greater rates of mineral sedimentation and labile C accumulation in the coastal plain estuary than in the coastal
lagoon (Unger et al., 2016). Potential sources of labile organic C include autochthonously produced plant biomass
and algae and allochthonous C deposited by tidal water. Several important questions arise from these findings
including: do high mineral sediment loads stimulate greater production and biomass of roots and rhizomes and is
belowground biomass (live and dead) a good indicator of C accumulation and accretion rates in marshes? For the
present study, we sought to examine how environmental conditions across marshes influence short-term process rates
of belowground productivity, decay, and longer term biomass stocks, and to relate these biotic processes and properties
to soil C and accretion rates. We used natural gradients in tidal elevation, sedimentation rates, salinity, and soil
properties in interior, short-form *Spartina alterniflora* marshes across two estuaries to test three overarching
hypotheses: (1) environmental parameters are highly correlated across marshes; however, hydrology is the most
important predictor of belowground productivity, decay rates, and above- and belowground biomass; (2) short-term
(< 2 yr) surface accretion rates are influenced by a combination of aboveground vegetation structure, belowground
productivity, decay and mineral sedimentation rate; and (3) longer-term (~50 years) accretion and soil C accumulation
are more strongly related to belowground biomass in organogenic marshes in a coastal lagoon than in more
minerogenic marshes of a coastal plain estuary, where the potential for allocthonous C contributions are greater.

**2 Methods**
**2.1 Study sites**




Six marshes in two estuaries along the Mid-Atlantic coast of the United States, the Delaware Bay and Barnegat Bay,
New Jersey were the focus of this study (Fig. 1). The two estuaries and marshes therein varied in geomorphic setting
and therefore, tidal range, hydrology, salinity, sediment availability, and nutrient concentrations also varied. The
Delaware Estuary is a large coastal plain estuary extending 215 km from the head-of-tides to the bay mouth at the
Atlantic Ocean. Tidal amplitude is approximately 1.5 m at the mouth and increases up-estuary modulated by estuary
and tidal channel geometry. Barnegat Bay is a shallow coastal lagoon extending 62.7 km along the coast of New
Jersey, separated from the Atlantic Ocean by a barrier island apart from two inlets. Barnegat Bay experiences a
relatively small tidal amplitude ranging from 20 to 50 cm depending on location in the bay (Defne and Ganju 2014).
Mean salinity ranges from 18 to 25 with lowest salinities in the northern part of the bay farther from the inlets and
near Toms River (Kennish 2001). Across the two estuaries, marshes ranged from being along a large tidal tributary
with relatively low salinity (Maurice River, MR, in Delaware Bay) to a back-barrier marsh (Island Beach State Park,
IB) in Barnegat Bay, which has been converting from *S. alterniflora* to shallow open water between the parallel
mosquito ditches over approximately the last decade, likely due to a lack of drainage through the ditches (pers obs).
Reedy Creek marsh in northern Barnegat Bay had interior ponds which have been expanding in size over time. These
systems have also been described in Elsey-Quirk 2016; Unger et al., 2016 and Boyd et al., 2017. Accretion rate in
Barnegat Bay marshes ($0.28 \pm 0.06$ cm/yr) over the last $50 - 100$ years was less than the rate of relative sea-level rise
over a similar time period (0.41 cm/yr; NOAA, Tides and Currents; *in* Boyd et al., 2017). In Delaware Bay salt
marshes accretion rate ($0.70 \pm 0.26$ cm/yr) exceeded the rate of local relative sea-level rise (0.34 cm/yr, NOAA, Tides
and Currents).

## 2.2 Experimental design

Five plot locations were established in each marsh along a transect of increasing distance from the marsh/estuary
boundary. Transects ranged from 270 to 2970 m with shorter transects in smaller marshes (generally in Barnegat Bay)
and longer transects in marshes along larger tidal channels (in Delaware Bay). This transect-based study design, rather
than replicate plots in close proximity, was employed to capture a range of within-marsh variation, while also
stratifying by short-form *S. alterniflora*. Belowground ingrowth and litterbag decomposition studies were employed
at each of the five locations (n = 5). Vegetation structure (i.e., stem density and shoot height and above- and
belowground biomass) was measured at three of the five locations (nearest, middle and farthest from the estuary).
Belowground biomass was determined from 2-cm depth sections from half of a 6-cm diameter soil core. The other
half of the core was used for analysis of soil properties, C (total organic, labile, and refractory) and nitrogen
concentrations, and $^{137}$Cs and $^{210}$Pb-activity (*in* Unger et al., 2016; Boyd et al., 2017). Two of the five locations in
each marsh (nearest and farthest) had continuous water level and salinity recorders installed. Hydrology was
extrapolated to the middle sample plot location using marsh elevation.

## 2.3 Environmental parameters

## 2.3.1 Hydrology and salinity



Two water level/conductivity recorders (In-Situ 5000 vented) were installed in each of the six marshes from August
2012 to October 2013.  Probes were placed in each marsh near and far from the estuary approximately 5 m from the
location of two of the five belowground ingrowth and litterbag plots and two of the three standing biomass and soil
core plots.  Probes were installed in slotted wells to a depth of 70 cm. Water level and conductivity were recorded
every 15 and 30 minutes, respectively.  Three GPS survey points were collected on the marsh surface around each
well and at each plot location using a Leica GS-14.

**2.3.2 Soil properties**
Soil cores were collected for testing relationships between belowground biomass and previously reported accretion
(*in* Boyd et al., 2017) and soil C accumulation rates (*in* Unger et al., 2016). PolyCate tubes 15.2 cm in diameter and
110 cm long with a sharpened bottom edge were used to collect soil cores. Cores were taken back to the lab and
sectioned into 2-cm depth sections.  Physical and chemical analyses were conducted on half of each the 2 cm sections.
Soil bulk density ($g/cm^3$), percent organic matter, total organic C, refractory and labile C, total nitrogen, and
radiometric dating were conducted and is reported in Unger et al., 2016 and Boyd et al., 2017.  The remaining half of
each 2-cm depth was rinsed and sieved for belowground biomass (see below).

**2.4 Vegetation parameters**

**2.4.1 Belowground ingrowth**
Belowground ingrowth rate (g DW $m^{-2}$ $yr^{-1}$) was measured at five locations in each of the six marshes using rates of
root and rhizome growth into flexible mesh bags filled with a 1:1 mixture of unfertilized top soil and peat moss
(McKee et al., 2007; n = 5). Ingrowth bags were constructed of flexible crawfish sack material with a mesh size of 6.3
x 3.2 mm, and, when filled, were 5 cm diameter and 15 cm length. Ingrowth typically underestimates absolute rates
of belowground productivity, but is useful for examining relative responses to varying environmental conditions
(Valiela et al., 1976; Graham and Mendelssohn 2016).  Four ingrowth bags were deployed in each of the five plots
and sequentially harvested approximately every four months from April 2013 to October 2014 to calculate
belowground (root + rhizome) ingrowth rate.  The four bags were deployed 50 cm apart in each plot and all bags were
installed vertically into 15 cm deep cored holes.  Accumulated root and rhizome mass was dried at 60°C to a constant
weight.  All of the roots and rhizomes in ingrowth bags were live based on color and structural integrity.

**2.4.2 Belowground decay**
Belowground decomposition of macro-organic matter was estimated using the litterbag technique.  Litterbags (20 x
10 cm) made of 1 x 1 mm window screen mesh were filled with 9 g wet weight of macro-organic material comprised
of coarse roots and rhizomes in a live to dead ratio of 1:3.  Five replicate samples were dried at 60°C to determine
initial dry weight.  Four litterbags were deployed horizontally 10-cm below the surface in each of five plots
approximately 2 m from ingrowth bags in each of the six marshes (n = 5). Litterbags in each plot were collected
sequentially over the same ~20 month period of ingrowth study for determination of mass loss over time.



Decomposition rates were calculated by fitting mass loss to the single negative exponential model with an asymptote
where a fixed proportion of recalcitrant organic material remains,
$$e^{-kt} = \frac{X_t}{X_0} - a$$

where $k$ is the decay constant, $x_t/x_o$ is the proportion of original mass remaining at time t, t is the time elapsed in
years, and a is the asymptote (Cotrufo et al., 2009).

**2.5.3 Biomass**

Aboveground biomass was harvested from three plots within a 0.25 $m^2$ quadrat where soil cores for radiometric dating,
C analysis and belowground biomass were also collected.  In the lab, stems were rinsed of mineral matter, counted,
and measured for height. Belowground biomass from half of each 2 cm soil section was rinsed to remove all mineral
sediment and separated into size classes of coarse and fine organic material. Coarse organic matter, comprised
primarily of stem bases and rhizomes, were further sorted into live and dead based on color and rigidity. A sieve with
a 2 mm mesh size retained coarse material and a sieve with a 1 mm mesh size retained fine organic matter, which
could not be easily separated into live and dead components. All above- and belowground biomass was dried to a
constant weight in a 70°C drying oven. Dry weights were converted to a g $m^{-2}$ basis and depth profiles were
constructed.

**2.5 Short-term accretion and organic accumulation rates**
Surface accretion and organic matter accumulation rates measured over the study period (Spring 2013 – Fall 2014)
were used to examine relationships with belowground ingrowth and decay rates.  Accretion rates measured from the
changes in height above a feldspar marker horizon can be highly temporally and spatially variable, and therefore, we
used a time period that was longer than belowground productivity and decay studies to increase the potential for a
significant linear trend. Three feldspar marker horizons were installed in 50 x 50 cm plots established approximately
5 m from standing biomass core locations in each marsh (n = 9).  A single plug was collected from each plot quarterly
and accretion was determined by the height of the marsh surface above the white feldspar marker (mm). A subsample
of surface soil was taken to the lab for measurement of wet and dry weight, and loss on ignition (LOI) at 550°C for 4
hours.  Annual accretion rate was calculated using the slope of the regression line for each plot and multiplying by
365. Organic accumulation rate was calculated from the product of bulk density of the surface soil (g/$cm^3$) determined
by the porosity, percent organic matter from LOI, and accretion rate (cm/yr). Mineral sediment accumulation rate was
also calculated using percent inorganic matter from sediment remaining following LOI.

**2.6 Data analysis**
Each environmental and vegetation parameter was checked for normality using the Shapiro-Wilks test. Log
transformations were used for most variables when necessary and appropriate and logit transformations were used for
percent data (e.g., percent soil organic matter, total nitrogen). A square root transformation was necessary to normalize





belowground ingrowth data. Matlab was used to calculate hydrologic parameters from the continuous water level time
series (MATLAB 6.1, the Mathworks Inc., Natick, MA). We used a nested analysis of variance to test for differences
among marshes nested within estuaries in belowground ingrowth, decay rate, and vegetation structure (e.g., stem
density, height, rooting depth, biomass). Multivariate correlation analysis was conducted to test for collinearity among
environmental predictor variables. One representative of highly correlated variables was chosen and redundant
variables were removed for future analyses. Correlations within environmental and vegetation data were expected,
and therefore, a multivariate approach were used to analyze the data. Multivariate analysis of variance (MANOVA)
was used to test for differences among marshes nested within estuaries in environmental parameters. If a significant
multivariate treatment affect was found based on the Wilks Lambda test, univariate tests were performed. Univariate
post-hoc tests were conducted using Tukey's HSD test. To examine the relationship between environmental and
vegetation parameters across samples, a stepwise regression was used with forward selection, starting with the full
model and minimum BIC. Belowground ingrowth, decay and biomass components were analyzed in separate stepwise
univariate models. Non-linear modelling was used when relationships were non-linear. Lastly, to test for relationships
between belowground biomass structure and accretion and C (total organic, labile and refractory) accumulation rates
a stepwise regression analysis was conducted. Unless otherwise specified, JMP V.12.1 was used for all statistical
analysis (JMP Version 12.1, SAS Institute Inc.).

**3 Results**

Elevations of marsh study locations ranged from -7 to 87 cm, referenced to the North American Vertical Datum 1988
(NAVD88) and were generally lower in Barnegat Bay than in Delaware Bay (Fig. 2). Relative to the tidal frame,
average elevations ranged from below mean low water (MLW; IB) to above mean high water (MHW; CC), with both
of these marshes in Barnegat Bay. Marsh elevation was more variable in Delaware Bay, but six of the nine locations
fell between mean low and high water. Overall, 65% of marsh areas had elevations within 5 cm of MLW, indicating
that the majority of the root zone was continuously inundated. The lowest MLW of the 18 study areas was 11 cm
below the surface. Tidal amplitudes ranged from 5 to 22 cm (Table 1). Barnegat Bay marshes were flooded for more
time ($53 \pm 12\%$) than Delaware Bay marshes ($40 \pm 9\%$; Estuary: p = 0.0341). However, there were also significant
differences among marshes (Marsh[Estuary]: $F_{4, 11} = 53.15$, p < 0.0001; Table 1). For example, in Barnegat Bay, time
flooded ranged from an average of 86% in IB to 6% in CC. Similarly, in Delaware Bay, DN was flooded 68% of the
time while DV was flooded only 11% of the time (Table 1).
Salinities ranged from 7 at MR in Delaware Bay to 40 at CC in Barnegat Bay. Salinity of Barnegat Bay marshes was
an average of 16 psu higher than DB, but there was also a significant difference within Barnegat Bay with RC in the
north having a lower salinity than IB and CC (Marsh[Estuary], p = 0.0103; Table 1).

**3.1 Multivariate correlations among environmental variables**

Correlation analysis of environmental factors across estuaries revealed two general groups of correlated variables
related to either hydrology or soil (Table 2). As might be expected, there were strong positive correlations among
hydrologic parameters. MHW was related to the greatest number of other hydrologic variables including % time



flooded, mean water level (MWL), MLW, and tidal range. Tidal range was positively related to MHW, but unrelated
to MLW as the average high water largely determined the tidal range experienced in the marsh and low water was less
variable with relatively little drainage in the marsh interior. Similarly, there were significant correlations among soil
parameters, with strong relationships among soil N content, bulk density and long-term mineral sedimentation rate.
Across marshes, bulk density and [137]Cs-based mineral sedimentation rate were positively correlated with each other
and negatively correlated with soil N content and soil organic matter.

When we conducted a correlation analysis within estuaries, different relationships emerged (correlations not shown).
In Barnegat Bay, hydrologic properties were strongly correlated with soil properties and processes. For example,
MHW was negatively related to mineral sedimentation rate, indicating that lower elevation marshes had less
sedimentation than higher elevation marshes. Here, localized sediment supply rather than flooding time influenced
sedimentation rates and, ultimately, marsh elevation. In Delaware Bay, hydrology and soil parameters were unrelated.
In both estuaries, salinity was negatively related to metrics of flooding (e.g., MHW, # flooding events, % time flooded)
indicating that marshes higher in the tidal frame had higher salinities.

Based on the results of the correlation analysis, two hydrologic parameters, MHW and MLW, were selected to
represent the suite of variables with which they were related, for which isolating individual relationships with
vegetation parameters would be impossible (Table 2).  Salinity was also maintained in the models, although it was
negatively correlated with several hydrologic parameters, but not related to soil properties. Because of the strong
relationship between soil bulk density and long-term mineral sedimentation rate, only sedimentation rate was retained
in subsequent models.

**3.2 Short-term processes**
**3.2.1 Belowground ingrowth**
Belowground growth of roots and rhizomes into ingrowth bags ranged from 0 – 550 g dw/m$^2$/yr.  Ingrowth rates were
significantly greater in MR and CC than in IB where 3 of the 5 locations had no root growth, little to no live
aboveground vegetation, and standing water for most of the year (Table 1; Fig. 3).  As we hypothesized, of all
environmental factors tested, only MLW influenced ingrowth rates (Fig. 4). This relationship was driven by extremes
of the range where locations with MLW ≥8 cm below the marsh surface, ingrowth was greater than 200 g/m$^2$/yr and
in IB where MLW was over 3 cm above the surface, root growth was zero (Fig. 4).

**3.2.2 Belowground decay**
While the percentage of root and rhizome material remaining at the end of approximately 20 months was similar
among marshes averaging 59 ± 1%, the rate of decay was greater in Barnegat Bay marshes than the Delaware Bay
marshes (Fig. 3).  We hypothesized that decomposition rate may not be related to measured environmental conditions;
however, we found a positive relationship between decay rate and water table salinity (Fig. 4).  Although stepwise
regression indicated that only salinity was significantly related to decay, three hydrologic variables were negatively



correlated with salinity. Therefore, it is possible that an increase in decay may also be promoted by a lower tidal range,
lower MHW, and/or fewer flooding events.

### 3.2.3 Surface accretion and organic accumulation rates

Short-term surface accretion rate was over two times greater in Delaware Bay marshes ($0.65 \pm 0.17$ cm/yr) than in
Barnegat marshes ($0.30 \pm 0.12$ cm/yr), largely driven by high accretion rates in MR and low accretion rates in IB (Fig.
3). Organic accumulation rate was significantly higher in MR and RC than in IB (Marsh[Estuary]: $F_{4,11} = 5.0$, p =
0.0155) and mineral accumulation rates did not significantly differ among sites or between estuaries over the time
period studied. Surface accretion rate was positively related to both organic and mineral sediment inventories above
the marker horizon, although was accretion was much more strongly influenced by mineral sediment than by organic
accumulation (Fig. 5). Neither aboveground vegetation structure (i.e., stem density, height, and biomass) nor
belowground processes (i.e., ingrowth and decay rates) were not related to surface accretion and accumulation rates.
Hydrology was the only environmental variable that influenced accretion and accumulation rates. Rates were
positively related to MHW and negatively related to MLW (Supplemental Table A), indicating that surface accretion
and accumulation rates increased in areas with a higher water above the marsh surface and greater drainage.

### 3.3 Vegetation structure

#### 3.3.1 Aboveground

Within marsh variability in *S. alterniflora* stem density was relatively high with coefficients of variation above 30%
for 5 of the 6 marshes and up to 155% (Table 3). Stem density in IB in particular increased with increasing distance
from the Bay ranging from zero stems approximately 67 m from the Bay, 102 stems/m$^2$ at 150 m distance, and 1372
stems/m$^2$ at 225 m landward. CC, the highest elevation marsh in Barnegat, had some of the highest stem densities (up
to 4112 stems/m$^2$). Shoot height was approximately two times greater in MR than the other marshes with the exception
of RC (Marsh[Estuary]: $F_{1,4} = 8.15$, P = 0.0026). Aboveground biomass was significantly related to both shoot height
and stem density as described by the equation: *AB = - 2032 + 521(log height) + 136(log density)* ($R^2 = 0.65$, p =
0.0030). Aboveground biomass was over six times greater in MR in Delaware Bay than RC and IB in Barnegat Bay
(Marsh[Estuary]: $F_{1,4} = 8.13$, P = 0.0021; Table 3).

#### 3.3.1 Belowground

Aboveground live biomass was not related to any component of belowground biomass, nor total belowground
biomass. Belowground biomass components did not scale with each other either. The 95% rooting depth was greatest
in DN, followed by MR, both of which had greater live root depths than Barnegat Bay marshes (Marsh[Estuary]: $F_{4,12} = 10.58$, p = 0.0007; Table 3; Fig. 6).

Belowground biomass was examined with both a temporal and spatial perspective. Our temporal examination
involved summing the biomass fractions relative to the $^{137}$Cs peak, which contributes to the variation in accretion rates
among marshes. However, biomass is continually added to the soil column, often to depths below the year 1963 $^{137}$Cs



marker, and therefore, a specific depth, 50 cm, was also used, which is typical methodology for quantifying
belowground biomass.  Live coarse belowground biomass measured to the $^{137}$Cs peak, which varied by 42 cm among
locations, was highly variable and did not differ significantly among marshes or between estuaries (Table 3). Live
coarse biomass ranged from an average of 505 g/m$^2$ in RC to 2675 g/m$^2$ in CC where the $^{137}$Cs peaks occurred at an
average of 11 and 16 cm depth, respectively.  In addition, the quantity of live biomass in 16 cm at CC was similar to
the amount of live biomass in MR where the $^{137}$Cs peak depth averaged 48 cm. However, much of the live biomass
was confined to the upper 30 cm in general, and therefore the difference in live biomass among marshes down to 50
cm depth was not substantially different from that measured to the respective $^{137}$Cs peaks.  Larger differences between
the depth-dependent calculations were found for the dead coarse and fine biomass. For example, marshes of Barnegat
Bay had $^{137}$Cs peaks that were 9 – 17 cm below the surface and the amount of dead coarse and fine biomass increased
by several times when depths to 50 cm were included. In contrast, there was less of an increase in Delaware Bay
marshes, where accretion rates were higher.  Dead coarse biomass was also highly variable and did not differ among
marshes using either calculation method.  Fine biomass in CC was greater than in RC when measured to the $^{137}$Cs
peak and greater than MR when measured to 50 cm depth (Table 3).  Total belowground biomass above the $^{137}$Cs peak
was significantly lower in RC than in DN and CC (Marsh[Estuary]: $F_{4,\,12}$ = 5.12, p = 0.0121). No significant difference
was found among marshes in total belowground biomass to a 50 cm depth.

Total live biomass (above-and belowground) was over three times greater in CC (3245 ± 768 g/m$^2$) than in RC (833
± 41 g/m$^2$) with no other differences among marshes (Marsh[Estuary]: $F_{1,4}$ = 4.2, P = 0.0227).  Live root-to-shoot ratio
ranged from 4 to 12 in marshes where live stems were present in all plots (Table 3).  Rates of ingrowth and decay
were not related to belowground biomass stocks.

**3.3 Environmental controls on vegetation structure**
While the aim of these analyses were to examine cross-system relationships between environmental conditions and
vegetation patterns, it became apparent that the responses of vegetation structure to environmental conditions were
highly estuary-dependent.  For example, live above- and belowground biomass were significantly related to long-term
rate of mineral sedimentation in Barnegat Bay only (Fig. 7a). Live aboveground biomass increased linearly with
increasing sedimentation rate and decreased linearly with increasing MHW, which were highly correlated in Barnegat
Bay (Fig. 7b).  This indicates that in Barnegat Bay, aboveground biomass responded positively to higher mineral
sedimentation and less flooding, and further that marsh areas experiencing higher MHW are suboptimal within their
elevational growth range. In Delaware Bay, aboveground biomass increased with increasing MHW following a
quadratic relationship (Fig. 7b).  Similar to aboveground biomass, live belowground coarse biomass, comprised of
stem bases, rhizomes and large roots, in Barnegat Bay was positively related to sedimentation rate and negatively to
MHW, while there were no significant relationships between live coarse biomass and measured environmental
parameters in Delaware Bay (Fig. 7c,d).  Dead belowground coarse biomass was also negatively related to MHW in
Barnegat Bay. Patterns of fine belowground biomass were not explained by environmental predictor variables in
Barnegat Bay; however, in Delaware Bay, fine biomass increased with decreasing mineral sediment accumulation rate





and increased with increasing salinity (Fig. 7f). Total belowground biomass was not explained by environmental
parameters in Barnegat Bay, but was negatively related to sedimentation rate in the Delaware Estuary.  Fine material
comprised 68 – 80% of the total belowground biomass in Barnegat Bay and 45 – 69% of total belowground biomass
in Delaware Bay.  High mineral sediment deposition in MR in Delaware Bay was associated with lower total
belowground biomass, primarily associated with less fine organic material.

**3.4 Relationship between belowground biomass structure and soil C and accretion**
Belowground biomass profiles corresponded well with profiles of organic C density, depending on the biomass
component (live coarse, dead coarse, or fine) and whether the C was chemically labile or refractory (Fig. 6). Labile C
density was related to live coarse biomass in 4 of the 6 marshes (Table 4).  Labile C density was not related to any
biomass fraction in IB and was related to only fine biomass in MR. Fine biomass was also related to refractory C
density in both MR and DV in Delaware Bay (Table 4).  In Barnegat Bay, refractory C was associated with all or most
of the belowground biomass fractions.
The pattern of total and live biomass was similar to that of total organic C accumulation in Barnegat Bay, but not in
Delaware Bay (Fig. 8).  Carbon accumulation and accretion rates were positively related to belowground biomass in
Barnegat Bay marshes, but not in Delaware Bay (Fig. 9).  Live coarse and fine biomass explained 86% of the variation
in OC accumulation in Barnegat Bay and was not significantly related to OC accumulation in DB (Fig. 9). When we
examined which structural components were related to organic C fractions, live coarse biomass was positively related
to labile C accumulation and fine biomass contributed to the accumulation of refractory C (Fig. 9). Biomass was not
related to accretion rate in Delaware Bay, but live and dead coarse biomass explained 78% of the variation in accretion
rate in Barnegat Bay (Fig. 9).

**4.0 Discussion**

Study locations in both estuaries spanned the tidal frame with *S. alterniflora* above, within, and below the average
tidal range. This relatively wide range of elevations and associated hydrologies along with other environmental
differences were predicted to be biologically significant. Our findings illustrate that within marsh variability in both
environmental conditions and vegetation response can be high such that marshes in very different geomorphological
settings can experience very similar abiotic conditions, and subsequent vegetation properties. Regardless of marsh
location, we found significant relationships between vegetation, decay and environmental factors. Belowground
ingrowth was negatively related to MLW, decay was positively related to salinity, and environmental-biomass
relationships were estuary-dependent. In Barnegat Bay, less flooding and higher sedimentation rates favored above-
and belowground biomass, which, in turn, increased soil C accumulation and accretion rates. Biomass dynamics in
Delaware Bay for the most part, were unrelated to environmental parameters, and had little influence on the
accumulation of soil C or accretion rate.




### 4.1 Geomorphic influences

Hydrology is predicted to be of primary importance to plant productivity and organic matter accumulation in marshes. Many experimental and modeling studies suggest that root growth and soil C production is maximized at an optimal water depth following a parabolic relationship with elevation (Morris et al., 2002; Kirwan and Guntenspergen 2012) and that the magnitude of organic production depends largely on elevation of the marsh (e.g., Kirwan et al., 2010; Mariotti & Fagherazzi 2010). While this theory may hold for gradients of inundation where sediment supply is constant and deposition is controlled by flooding time, at a larger spatial scale, inundation and sediment supply can vary in non-uniformly (e.g., Butzeck et al. 2015). Belowground responses to flooding have been suggested to be independent of mineral sediment availability (Kirwan and Guntenspergen 2012). However, this conclusion was based on a study in a sediment-limited marsh system where only the effect of flooding and associated sediment deposition was altered. The distinction between flooding and associated sediment deposition and spatial variation in sediment availability is an important one. Our study illustrated that above- and belowground biomass was positively influenced by elevation and a high long-term sediment deposition rate in marshes of a coastal lagoon where sediment supply can be localized and estuarine tidal dynamics can limit the transport and deposition of sediments, and therefore, marshes are highly dependent on organic matter production for accretion. Here, a high sediment supply promoted a positive feedback between belowground biomass and elevation. However, at similar rates of mineral sedimentation, belowground live coarse biomass was over 1000 $g/m^2$ lower in the coastal plain estuary, potentially associated with higher soil redox potential and a lower requirement for rhizome photosynthate storage (Gallagher and Kibby 1981). In marshes of the coastal plain estuary, live belowground biomass was unrelated to measured environmental parameters, and biomass was not related to rates of soil C accumulation or accretion.

To examine the relative influence of hydrology and mineral sedimentation on plant productivity, decay and biomass, it is useful to compare two marshes where the hydrologic conditions in the marsh interior were similar, RC in Barnegat Bay and DN in Delaware Bay. Tidal range nearest RC and DN averaged approximately 0.3 m and 1.5 m, respectively (USGS gage 01408167 and 01411435, respectively). In the marsh interior, however, water level statistics between marshes were quite similar. For example, the average time flooded was approximately 67% at both sites, and MHW, MLW, and tidal range were all comparable. Although not significantly different, the frequency of flooding was higher and the duration of flooding was slightly lower in RC than in DN. Salinity also averaged about 7 psu greater in RC, although not significantly different from that in DN. Given these relative similarities in hydrology, we might predict similar rates of plant productivity and decay and quantities of biomass. While short-term process rates were similar between these two marshes, biomass and longer-term processes were more different, and much higher in DN than in RC. Aboveground biomass averaged two times higher in DN than RC. Belowground live coarse biomass averaged 2456 and 721 $g/m^2$ in DN and RC, respectively and dead coarse biomass averaged 1063 and 280 $g/m^2$ in DN and RC, respectively. Importantly, these two biomass components were most significantly related to long-term accretion rate in Barnegat Bay, and $^{137}Cs$ accretion rates were two and a half times greater in DN than RC (Boyd et al., 2017). Live rooting depth was also significantly greater in DN (46 cm) as compared to RC (19 cm). Despite similar hydrology in the marsh interior, mineral sedimentation rates were an order of magnitude larger, and soil bulk density was three




times greater in DN than in RC (Boyd et al., 2017; Unger et al., 2016).  In Barnegat Bay, differences in mineral
sedimentation explained 62% of the variation in live belowground biomass and RC had among the lowest
sedimentation rate of all the marshes in the study.  Geomorphic conditions of high sediment availability and larger
tidal range in the estuary and creeks, which can mobilize and distribute sediments likely work in concert to promote
greater long-term deposition rates in DN than in RC.  To further illustrate the importance of mineral sedimentation to
belowground biomass, we can examine CC and DV, two of highest elevation marshes in the two estuaries.  DV was
high in elevation closer to Delaware Bay and was lower further inland, resulting in some variability with respect to
hydrology.  Overall, many of the hydrologic parameters were comparable between CC and DV.  Salinity, however,
was significantly higher at CC than DV, but soil organic matter, bulk density and mineral sedimentation rate were
similar.  Belowground ingrowth, and surface accretion rates were also similar, and although decay rates were faster in
CC, potentially related to higher salinity, the percentage of mass remaining in litterbags was similar to DV.
Belowground biomass and biomass profiles were also similar between marshes with the exception of greater fine
organic matter in CC, particularly between 4 and 20 cm depth.  With both hydrology and sedimentation rates being
within range of each other, these two marshes in very different geomorphic settings, had comparable biomass and
organic C accumulation rates.

### 4.2 Belowground ingrowth

Belowground ingrowth reflects root growth into new sediment in absence of space-limitation by other roots, rhizomes
and organic matter in natural marsh soils.  The experimental increase in resource space inherent to the ingrowth
methodology can result in belowground responses to abiotic conditions that may differ from that of biomass.  For
example, nutrient enrichment led to an increase in belowground ingrowth while resulting in a decline in live
belowground biomass (Graham and Mendelssohn 2015).  Inundation has been shown to have varying effects on root
ingrowth depending on species (Janousek et al., 2016), but ultimately root productivity declines as inundation time
exceeds the tolerance limit of the species. *Spartina alterniflora* is highly adapted to flooded conditions, possessing
both morphological adaptations such as aerenchyma to facilitate oxygen transport to flooded rhizomes and roots and
physiological adaptations such as anaerobic metabolism (Teal and Kanwisher 1966; Mendelssohn et al., 1981).  We
observed that root growth of *S. alterniflora* was highly variable under moderate flooding conditions, potentially more
affected by other abiotic conditions or the density of parental live root structures in close proximity. Because mean
low water was generally above the lowest part of the ingrowth bag, and ingrowth bags were surrounded by water most
of the time, the effect of inundation on root growth was evident at the extremes of MLW, where either average low
water level exposed a significant portion of the ingrowth bag resulting in relatively high root growth or low water
averaged above the surface and the root zone was continuously flooded, where no roots grew.  Thus, as others have
found (Blum 1993; Kirwan and Guntenspergen 2012), when new substrate is available, root growth increased with
drainage.

### 4.3 Belowground decay





Environmental effects on belowground organic matter decomposition in marshes have been somewhat inconclusive.
Continuously moist organic substrates (Valiela et al., 1984), high microsite variability in root oxidation (Teal and
Kanwisher 1961) and contact with live roots (Hackney and de la Cruz 1980) create conditions where a seemingly
homogeneous marsh can high variability in decomposition (Hackney 1987). Some studies have shown that decay
rates are not affected when placed in environments with different flooding regimes and redox potentials (Valiela et
al., 1984; Blum 1993), while other studies have shown C dioxide emissions to be greater with a lower water table
depth (Nyman and DeLaune 1991). Depth-dependent decay suggests that decomposition slows in more anaerobic,
potentially cooler, conditions deeper in marsh sediments (Hackney and de la Cruz 1980; Hemminga et al., 1988). In
mesocosms, the relationship between belowground decay and flooding was site-specific where decay increased with
increasing inundation at two of the three study locations (Kirwan et al., 2013). Contrasts in the effects of inundation
have been found between experimental mesocosms and natural marshes, where decomposition of roots and rhizomes
increased with increasing inundation in mesocosms, but in the field decomposition rates were higher in the high as
compared to the low marsh (Janousek et al., 2017). The difference between two outcomes was hypothesized to be
related to other abiotic factors varying along the tidal gradient or differences in nutrient availability and soil organic
matter between mesocosm and marsh. Hemminga et al. (1988) found an increase in the belowground decomposition
of *Spartina anglica* with increased elevation, potentially associated with less inundation time, higher temperatures,
and/or higher salinity. Some have found negative (Hemminga et al., 1991) or no effect (Connolly et al., 2014) of
salinity on marsh plant leaf litter decay at salinities of $\leq 25$. We found a significant positive relationship between
belowground decay rate and salinity, which was negatively correlated with tidal range, MHW, and number of flooding
events per year. Ultimately, the more exposure and higher salinity, the greater the rate of decomposition. Salinity in
the present study ranged from an average of 7 to 39. Along a lower gradient in salinity ($\sim 0 - 15$) in *Phragmites*
*austrailis*- dominated marshes, belowground decay of cotton strip was related more to nutrient availability and specific
constituents in seawater (e.g., boron, calcium, and sulfur) rather than the major seawater cations or salinity as a whole
(Mendelssohn et al., 1999). Lower organic matter preservation along estuary salinity gradients from tidal freshwater
to saline marshes has been suggested to be associated with a positive effect of salt water on decomposition (Craft,
2007). While decay rates were higher in the more saline marshes of Barnegat Bay than Delaware Bay marshes, there
was essentially no difference in the percent remaining at the end of approximately 20 months, which averaged 59%.
The implications are that regardless of environmental conditions, approximately 60% of the plant-derived organic
matter in the soil may be recalcitrant and undergo relatively slow decomposition. This percentage corresponds well
to measurements of lignocellulose, which comprises approximately 79.5% of root and rhizome biomass (Hodson et
al., 1984). The cellulosic portion undergoes higher rates of mineralization than lignin, which comprises approximately
19.3% in roots and rhizomes of short-form *S. alterniflora* (Hodson et al., 1984) resulting in some additional losses
over time.

**4.4 Biomass**
Belowground biomass of *S. alterniflora* can also be quite variable both temporally and spatially. Seasonal changes in
live roots and rhizomes can be due to growth of new tissues, turnover, and/or translocation of non-structural




carbohydrates to aboveground tissues (Gallagher 1983). Decay and turnover rates influence changes in the dead
biomass. Thus, biomass-environment relationships can change throughout the year and between years (Reed and
Cahoon 1992). High micro-spatial variability can also confound measurements of seasonal and inter-annual biomass
variations. Spatially, elevation is predicted to be a primary determinant of belowground biomass, as aboveground
biomass responds relatively quickly to changes in inundation. However, it is yet unclear how above-and belowground
biomass scale with each other across complex environmental gradients. Across a gradient of elevation (> 10 cm) in a
marsh on Galveston Island, Texas, *S. alterniflora* aboveground live biomass decreased with increasing elevation, while
root biomass was similar among elevation ranges (Kulawardhana et al., 2015). In a subsiding marsh in Louisiana,
belowground biomass was positively related to marsh surface elevation (Reed and Cahoon 1992). Our data suggest
that above- and belowground biomass do not scale with each other and that belowground live biomass positively
responds to sedimentation when geomorphic conditions limit tidal range and sediment availability. In the coastal plain
estuary, where mineral sediment accumulation rates ranged from less than 500 to over 4000 $g/m^2/yr$, belowground
biomass did not vary much along this gradient and, for the most part, was not related to measured environmental
explanatory variables.

### 4.5 Relationship between vegetation and soil C and accretion

Positive relationships between *S. alterniflora* belowground productivity and biomass and soil C, C accumulation and
accretion in salt marshes have largely been assumed by high plant production estimates, observations of plant derived
biomass in soil cores and positive correlations between organic matter accumulation and accretion. However, few, if
any, studies have directly tested these relationships across a natural gradient of geomorphology. Over short time
scales (≤ 2 yrs), surface accretion was positively related to mineral sedimentation rate, and to a lesser extent organic
matter accumulation rate. We found no significant relationships between root ingrowth, decay and surface accretion
rate, likely as other factors influenced surface accretion such as sediment deposition and, potentially the accumulation
of surface litter. We found that geomorphology played a large role in influencing the relative importance and
contribution of plant biomass to soil C and marsh accretion rate. Soil C density and biomass profiles illustrated varying
relationships between biomass structural components and labile and refractory C. Two interesting relationships
emerged from the analysis. One is that in the organogenic marshes in Barnegat Bay, the refractory C is supported by
all or most biomass components including live biomass. This was not the case in Delaware Bay marshes, where the
refractory C was supported by either fine or dead coarse biomass. This finding supports the hypothesis that under
constrained growing conditions (e.g., high salinity, high inundation, low tidal range) found in Barnegat Bay marshes,
more energy may be invested into the production of recalcitrant tissues such as lignin, cellulose and hemicellulose. In
Delaware Bay, as we would predict, refractory C was associated with the dead biomass and, in most sites, the broken
down fine biomass. Interestingly, in the marsh with the highest accretion and mineral sediment accumulation rate and
the lowest fine biomass (MR), labile C density was only associated with fine biomass. This may be associated with
greater mineral protection of small labile tissues (as hypothesized in Unger et al., 2016).
While relationships between biomass and C densities were relatively strong for all marshes, C accumulation and
accretion rates were more strongly related to biomass in Barnegat Bay marshes than in Delaware Bay marshes. The





hypothesis that belowground biomass had a greater influence on soil C accumulation and accretion rates in
organogenic marshes than minerogenic marshes was supported in this study. Based on these data, belowground
biomass (live, dead, fine and/or total) is not necessarily a good predictor of carbon accumulation and accretion rates.
Although previously unknown, it may have been presumed that marshes with higher tidal ranges and higher rates of
mineral sedimentation would have higher stocks of belowground biomass. A lower MLW may promote the growth of
roots and rhizomes to a greater depth, and mineral sediment may create more accommodation space and provide more
mineral nutrients to support growth.  However, at high sedimentation rates, such as those observed in marshes of the
Delaware Bay, belowground biomass was largely uncoupled with rates of organic C accumulation, despite significant
correlations between biomass and C (labile and refractory) density.  At the low end of mineral sedimentation rates (<
1000 $g/m^2/yr$), sediment accumulation, regardless of local hydrologic constraints, was of primary importance in
influencing plant biomass, and C accumulation and accretion rates. Hydrology in the estuary and tidal channels is of
primary importance for mobilizing and distributing sediments onto the marsh surface.  Importantly, while studies
suggest the vulnerability of microtidal marshes in coastal lagoons (Reed et al.,  2008; Kirwan & Guntenspergen 2010;
Ganju et al.,  2017), we illustrate the importance of localized sources of sediments as being the key to their survival
through the positive feedback on live and dead coarse and fine root biomass, and organic matter and C accumulation
rates.  Continuous standing water, enlargement of interior ponds, and loss of aboveground vegetation are all signs of
marsh deterioration. At IB, two of the sampling locations had permanent shallow water between ditches, and had lost
the aboveground biomass. Remnants of former vegetation was evident belowground, where significant quantities of
live stem bases, roots and rhizomes were present to below 28 cm depth at least three years after aboveground biomass
was permanently lost. Therefore, belowground biomass is not always a good indicator of marsh health and
vulnerability.  In terms of soil C and accretion, our study indicates that biomass can be related to soil C accumulation
and accretion rates, but at the low end of a range in mineral sediment accumulation. At higher sedimentation rates,
fine and total belowground biomass declined significantly. The mechanism of fine organic matter loss or limited input
with greater mineral sedimentation rates is unclear, particularly when these marshes have the highest rates of labile
and total organic C accumulation in our study (Unger et al., 2016).  However, the preferential loss of fine organic
matter, which is largely recalcitrant, may have influenced the relatively high labile C content of the soil.

**4.6 Conclusions**
The fate of low-lying salt marshes as sea-level rises depends, in part, on their ability to accumulate organic matter and
to trap sediments.  Sediment supply is also a major factor, and may be most important in influencing the biophysical
processes that promote accretion and soil C accumulation.  Our study illustrates that while drainage enhances new
root growth into unoccupied substrate, biomass dynamics may be more strongly related to rates of mineral
sedimentation and availability of substrate in *Spartina alterniflora* marshes where sediment availability is limiting
promoting positive feedbacks between biomass, soil C accumulation, and elevation. Soil C and accretion were strongly
related to biomass fractions in the coastal lagoon estuary, where all biomass components were positively related to
refractory C.  When sediments are readily available, belowground biomass was not strongly related to important
environmental variables (e.g., hydrology, salinity, sedimentation rate), and was not related to soil C accumulation and



accretion rates, which were the highest in our study. In the coastal plain estuary, belowground biomass did not scale
with accretion and accumulation indicating other, smaller-sized organic material, potentially allocthonous, was
contributing to the high rates of C accumulation and accretion in these marshes.





*Acknowledgements* We would like to thank M. Archer, P. Zelanko, L. Zaoudeh, M. Schafer, M. Mills, M. Brannin,
and B. Boyd for help in the field and lab.  We would also like to thank all of the Mid-Atlantic Coastal Wetland
Assessment partners including D. Kreeger, M. Maxwell-Doyle, D.J. Velinsky, A. Padaletti et al., . This research was
largely funded by NJ SeaGrant / NOAA Grant #6210-0011. Surface accretion data were funded by EPA Region 2
Wetland Program Development Grant CD-97225909.


*Competing interests* The authors declare that they have no conflict of interest.




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





**Table 1:** Environmental conditions of interior *Spartina alterniflora*-dominated marshes in two estuaries of the Mid-Atlantic, USA. Values are means ± standard errors (n = 2 for hydrologic parameters and n = 3 for soil and elevation and soil parameters).

| Environmental parameter | Barnegat Bay | | | Delaware Bay | | |
|---|---|---|---|---|---|---|
| | RC | IB | CC | DV | MR | DN |
| Distance to tidal channel (m) | 13 ± 5a | 11 ± 3a | 19 ± 3a | 56 ± 16b | 27 ± 16b | 62 ± 34b |
| Marsh elevation (cm, NAVD88) | -2.3 ± 6.4 | 11.8 ± 1.9 | 29.1 ± 1.0 | 71.9 ± 7.7 | 51.5 ± 16.3 | 66.8 ± 7.7 |
| % time flooded | 66 ± 4ab | 86 ± 1a | 6 ± 1c | 11 ± 4c | 40 ± 1b | 68 ± 5ab |
| Mean high water (cm) | 12.6 ± 1.2ab | 10.3 ± 0.7bc | -3.3 ± 0.8d | 3.4 ± 3.7cd | 21.2 ± 1.8a | 15.6 ± 1.2ab |
| Mean water level (cm) | 3.5 ± 0.7 | 5.5 ± 1.0 | -6.5 ± 0.7 | -4.9 ± 1.9 | 2.7 ± 0.3 | 2.8 ± 0.5 |
| Mean low water (cm) | -0.8 ± 1.0a | 3.1 ± 1.1a | -8.5 ± 0.7b | -7.4 ± 1.6b | -1.0 ± 2.0a | -1.4 ± 0.7a |
| Tidal range (cm) | 13.4 ± 0.6bc | 7.2 ± 0.5cd | 5.1 ± 0.1d | 10.9 ± 2.2bcd | 22.2 ± 0.2a | 17.0 ± 1.9ab |
| # flooding events/yr | 304 ± 21ab | 324 ± 14a | 113 ± 16bc | 205 ± 51bc | 455 ± 55a | 176 ± 78bc |
| Avg duration of flood (hr) | 20 ± 2b | 24 ± 1c | 4 ± 1b | 15 ± 7b | 7 ± 1b | 44 ± 14b |
| Salinity | 20.6 ± 0.1bc | 30.4 ± 4.0ab | 38.8 ± 0.8a | 17.1 ± 1.0c | 7.4 ± 1.0c | 13.5 ± 2.1c |
| Soil nitrogen (%) | 1.6 ± 0.2a | 1.5 ± 0.1a | 0.8 ± 0.1b | 0.7 ± 0.2b | 0.5 ± 0.1b | 0.5 ± 0.1b |
| Soil organic matter (%) | 39.7 ± 1.3a | 39.4 ± 0.3a | 23.4 ± 1.4ab | 32.4 ± 9.1a | 14.1 ± 0.8b | 27.8 ± 5.5ab |
| Soil bulk density (g/cm3) | 0.14 ± 0.01c | 0.15 ± 0.02bc | 0.29 ± 0.01abc | 0.34 ± 0.07ab | 0.52 ± 0.04a | 0.39 ± 0.03a |
| Long-term mineral sedimentation rate (g/m2/yr)* | 147 ± 22d | 168 ± 19cd | 626 ± 45bc | 1237 ± 576ab | 4126 ± 74a | 1568 ± 279ab |




**Table 2: Multivariate correlations and associated correlation probabilities for environmental parameters in saline marshes of Delaware Bay and Barnegat Bay, NJ. Only parameters with significant correlations are shown. Properties related to hydrology and soils are have a blue and brown background, respectively.**

| Parameter | Marsh elevation | % time flooded | MHW (cm) | MWL (cm) | MLW (cm) | Tidal range (cm) | # flooding events/yr | LOG flood duration | Salinity | LOGIT soil N | LOGIT soil organic matter density | Soil bulk density | LOG long-term mineral sedimentation rate |
|---|---|---|---|---|---|---|---|---|---|---|---|---|---|
| Marsh elevation (cm, NAVD88) | 1.00 / <0.0001 | | | | | | | | | -0.85 / 0.0004 | | 0.78 / 0.0027 | 0.75 / 0.0046 |
| LOGIT % time flooded | 0.71 / 0.0089 | 1.00 / <0.0001 | 0.97 / <0.0001 | 0.94 / <0.0001 | | | | 0.84 / 0.0006 | | | | | |
| MHW (cm) | 0.97 / <0.0001 | 0.83 / <0.0001 | 1.00 / <0.0001 | 0.95 / <0.0001 | 0.94 / <0.0001 | 0.85 / <0.0001 | | 0.74 / 0.0007 | -0.69 / 0.0409 | | | | |
| MWL (cm) | 0.94 / <0.0001 | 0.74 / 0.0007 | 0.95 / <0.0001 | 1.00 / <0.0001 | 0.95 / <0.0001 | | | 0.74 / 0.0007 | | | | | |
| MLW (cm) | | 0.74 / 0.0007 | 0.94 / <0.0001 | 0.95 / <0.0001 | 1.00 / <0.0001 | | | 0.82 / <0.0001 | | | | | |
| Tidal range (cm) | | | 0.85 / <0.0001 | | | 1.00 / <0.0001 | 0.81 / <0.0001 | | -0.87 / 0.0023 | | | | |
| # flooding events/yr | | | | | | 0.81 / <0.0001 | 1.00 / <0.0001 | | -0.68 / 0.0428 | | | | |
| LOG flood duration (hr) | 0.84 / 0.0006 | 0.82 / <0.0001 | 0.74 / 0.0007 | 0.74 / 0.0007 | 0.82 / <0.0001 | | | 1.00 / <0.0001 | | | | | |
| Salinity | | | -0.69 / 0.0409 | | | -0.87 / 0.0023 | -0.68 / 0.0428 | | 1.00 / <0.0001 | | | | |
| LOGIT soil N (%) | -0.85 / 0.0004 | | | | | | | | | 1.00 / <0.0001 | 0.74 / 0.0054 | -0.93 / <0.0001 | -0.96 / <0.0001 |
| LOGIT soil organic matter (%) | | | | | | | | | | 0.74 / 0.0054 | 1.00 / <0.0001 | -0.78 / 0.0025 | -0.84 / 0.0005 |
| Soil bulk density (g/cm³) | 0.78 / 0.0027 | | | | | | | | | -0.93 / <0.0001 | -0.78 / 0.0025 | 1.00 / <0.0001 | 0.94 / <0.0001 |
| LOG mineral sedimentation rate (g/m²/yr) | 0.75 / 0.0046 | | | | | | | | | -0.96 / <0.0001 | -0.84 / 0.0005 | 0.94 / <0.0001 | 1.00 / <0.0001 |





**Table 3: Vegetation structure in *Spartina alterniflora* salt marshes in two Mid-Atlantic estuaries (n = 3, ± standard error).**

| | BB | | | DB | | |
|---|---|---|---|---|---|---|
| | RC | IB | CC | DV | MR | DN |
| Stem density (#/m²) | 509 ± 132 | 493 ± 440 | 3276 ± 615 | 1234 ± 382 | 612 ± 174 | 1675 ± 53 |
| Average height (cm) | 17 ± 3[ab] | 14 ± 2[b] | 13 ± 2[b] | 13 ± 1[b] | 26 ± 1[a] | 12 ± 1[b] |
| Aboveground biomass (g/m²) live | 112 ± 41[b] | 93 ± 88[b] | 362 ± 38[ab] | 234 ± 79[b] | 664 ± 90[a] | 353 ± 35[ab] |
| dead | 338 ± 129 | 91 ± 85 | 317 ± 97 | 33 ± 8 | 122 ± 38 | 100 ± 10 |
| 95% live rooting depth (cm) | 19 ± 1[c] | 20 ± 1[c] | 18 ± 1[c] | 23 ± 1[bc] | 30 ± 2[b] | 46 ± 6[a] |
| Belowground biomass (g/m²) *to ¹³⁷Cs-peak* | | | | | | |
| live coarse | 505 ± 21 | 1225 ± 200 | 2675 ± 764 | 1842 ± 7 | 2055 ± 700 | 1973 ± 201 |
| dead coarse | 138 ± 91 | 131 ± 69 | 310 ± 180 | 341 ± 191 | 985 ± 124 | 708 ± 469 |
| fine | 1498 ± 612[b] | 3676 ± 186[ab] | 4326 ± 258[a] | 3398 ± 438[ab] | 2484 ± 532[ab] | 3527 ± 448[ab] |
| *to 50 cm depth* | | | | | | |
| live coarse | 721 ± 81 | 1568 ± 222 | 2839 ± 758 | 1931 ± 34 | 2055 ± 700 | 2456 ± 305 |
| dead coarse | 280 ± 73 | 952 ± 231 | 1262 ± 125 | 690 ± 106 | 1010 ± 111 | 1063 ± 494 |
| fine | 4406 ± 1280[ab] | 8192 ± 2005[ab] | 8999 ± 948[a] | 6599 ± 1654[ab] | 2517 ± 565[b] | 5626 ± 661[ab] |
| Live BG:AB ratio | 8 ± 3 | 62 ± 56 | 8 ± 3 | 12 ± 5 | 4 ± 2 | 7 ± 1 |







**Table 4: Results of regression analysis of the relationship of belowground biomass to labile and refractory soil C density.**

| C density fraction (g/cm$^3$) | Marsh | Relationship with belowground biomass (g/m$^2$) |
|---|---|---|
| labile | RC | Live coarse: adj R$^2$ = 0.25, P < 0.0001 |
| | IB | ns |
| | CC | Live coarse: adj R$^2$ = 0.49, P < 0.0001 |
| | DV | Live coarse: adj R$^2$ = 0.24 , P < 0.0001 |
| | MR | Fine: adj R$^2$ = 0.60, P = 0.0172 |
| | DN | Live coarse: adj R$^2$ = 0.23 , P < 0.0001 |
| refractory | RC | Live and fine: adj R$^2$ = 0.60 , P < 0.0001 |
| | IB | ALL: adj R$^2$ = 0.64 , P < 0.0001 |
| | CC | ALL: adj R$^2$ = 0.70 , P < 0.0001 |
| | DV | Fine: adj R$^2$ = 0.17, P = 0.0001 |
| | MR | Fine: adj R$^2$ = 0.53, P = 0.0306 |
| | DN | Dead coarse: adj R$^2$ = 0.09, P = 0.0389 |





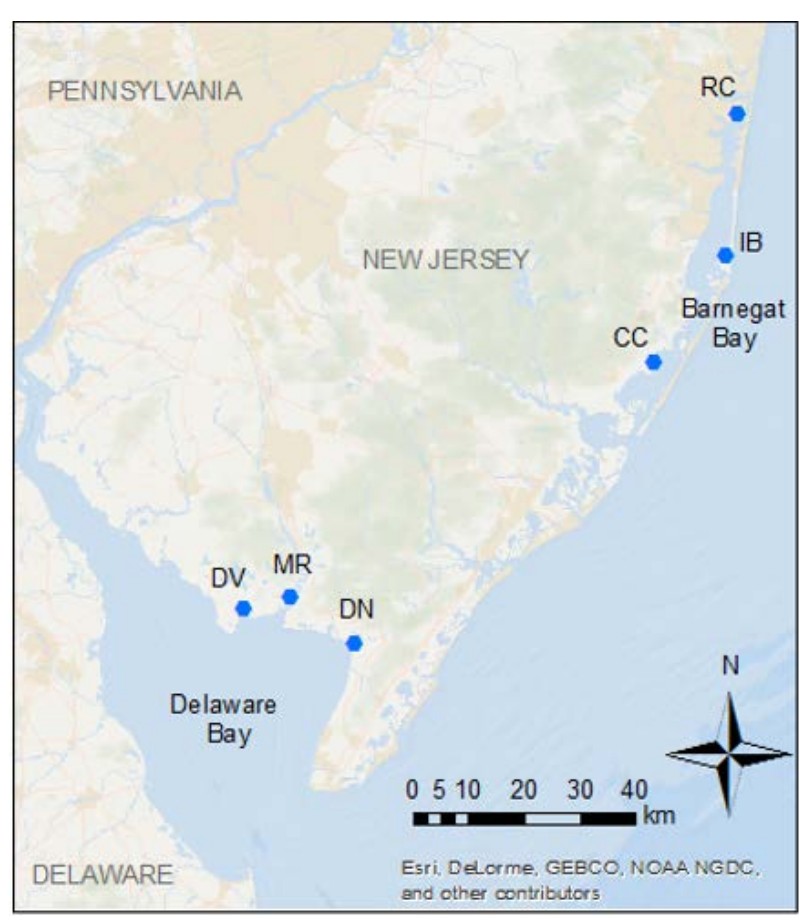

**Figure 1: Study locations in Barnegat Bay and Delaware Bay along the mid-Atlantic coast, U.S.**





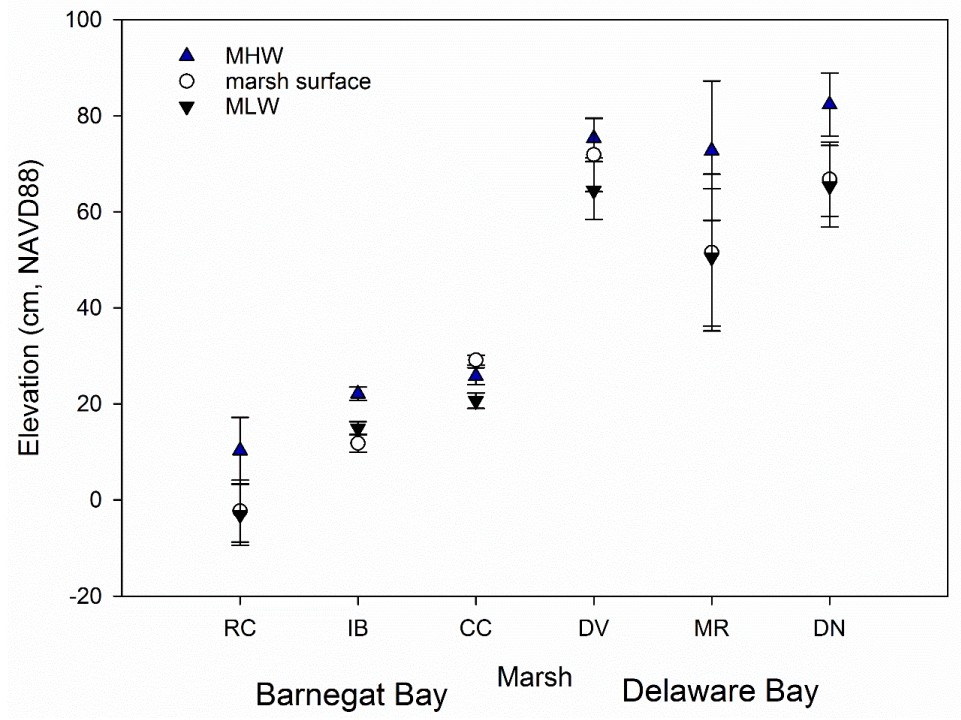

**Figure 2. Mean high and mean low water and marsh surface elevations based upon elevation survey data and**

**continuous water level recorded 8/12 – 10/13 in six *Spartina alterniflora* marshes in the mid-Atlantic, US (n =**

**2; 2 water level recorders/marsh).**





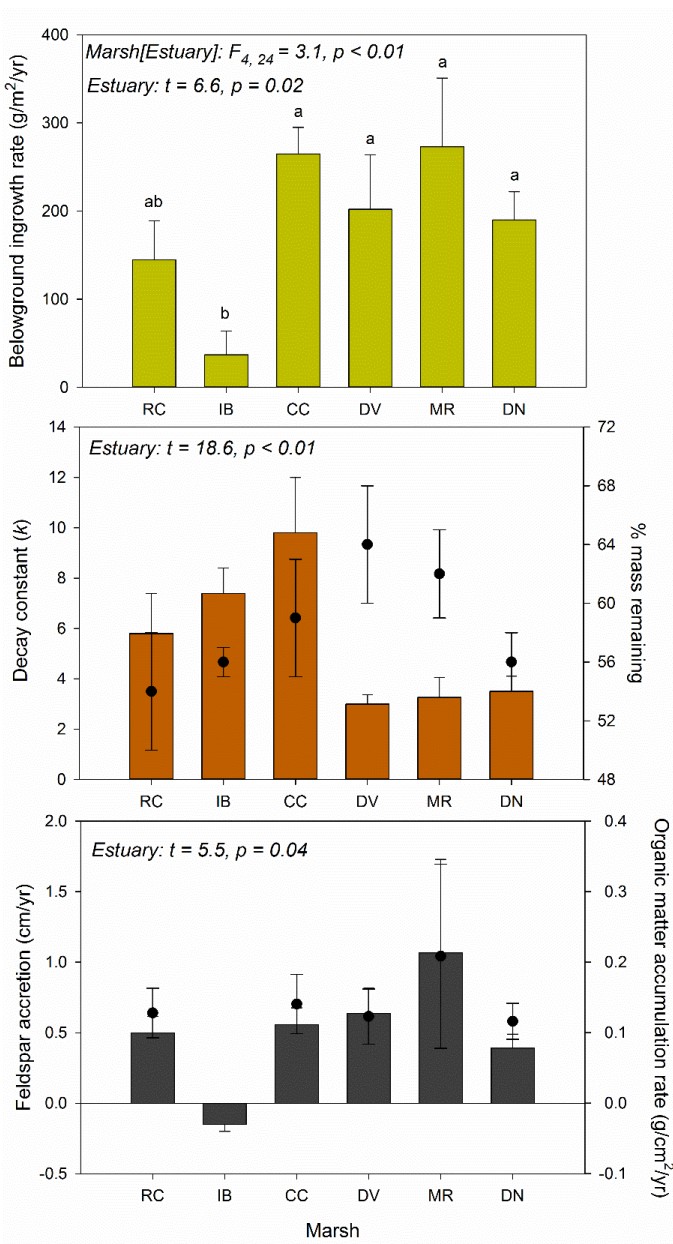

**Figure 3: Belowground ingrowth rate, decay constant and percent mass remaining, and short-term surface accretion and organic matter accumulation rates in salt marshes of two mid-Atlantic estuaries (n = 3, ± standard error). Decay rates are illustrated by vertical bars while percent mass remaining are dots. Similarly, accretion rate values are represented by bars and organic accumulation rates by dots. Different letters indicate significant differences (p < 0.05) with lower case letters corresponding to bar graphs and uppercase letters to scatter graphs.**



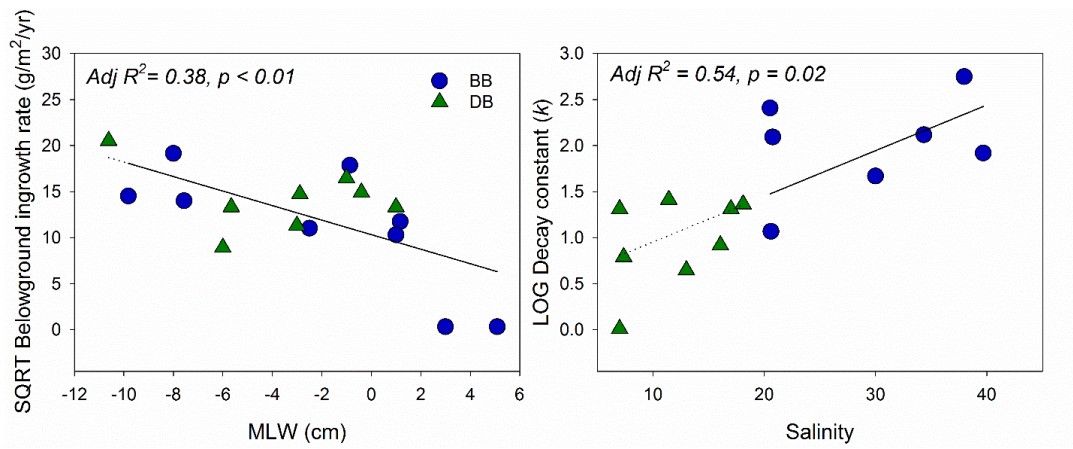

**Figure 4: Relationships between belowground ingrowth and mean low water (MLW) relative to the marsh surface (left) and decay constant from litterbags and salinity (right) based on the results of stepwise regression across Barnegat and Delaware bays. Blue dots and green triangles refer to data collected in Barnegat Bay (BB) and Delaware Bay (DB), respectively.**





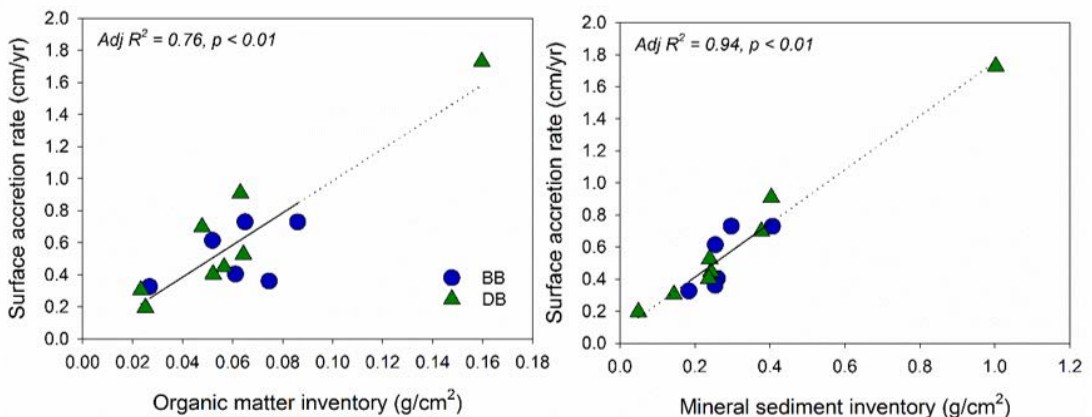

28 **Figure 5: Relationship between surface accretion and organic and mineral accumulation rates above feldspar**
29 **marker horizons in marshes of two estuaries in the Mid-Atlantic U.S. coast across Barnegat and Delaware bays.**
30 **Blue dots and green triangles refer to data collected in Barnegat Bay (BB) and Delaware Bay (DB), respectively.**



**Figure 6: Belowground biomass and labile and refractory organic C density profiles in marshes of two Mid-Atlantic estuaries. Biomass data are means (standard errors not shown). C density data previously reported in Unger et al., 2016.**



**Figure 7: Relationships between vegetation structure and environmental parameters. For these analyses, belowground biomass to 50-cm depth was used. Mineral sedimentation rates were calculated using $^{137}$Cs-dating, and therefore, are average rates over the last 50 years. Blue dots and green triangles refer to data collected in Barnegat Bay (BB) and Delaware Bay (DB), respectively, which were analysed separately. Both significant and non-significant (n.s.) relationships are shown.**






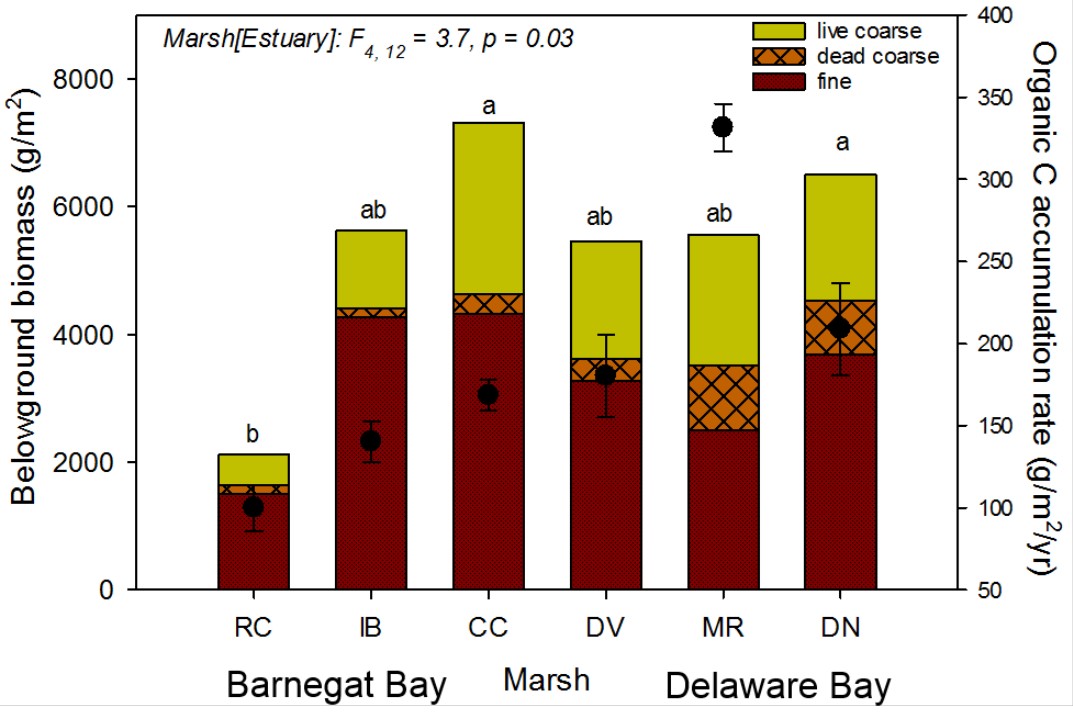

**Figure 8: Belowground biomass (scaled on left-axis) and organic C accumulation rate (scaled on right axis;**
**previously reported in Unger et al., 2016) in marshes of two Mid-Atlantic estuaries. Belowground biomass and**
**C accumulation are relative to the [137]Cs peak depth. Statistics for total belowground are shown and letters**
**represent differences (p < 0.05). Standard errors are shown in Table 2.**

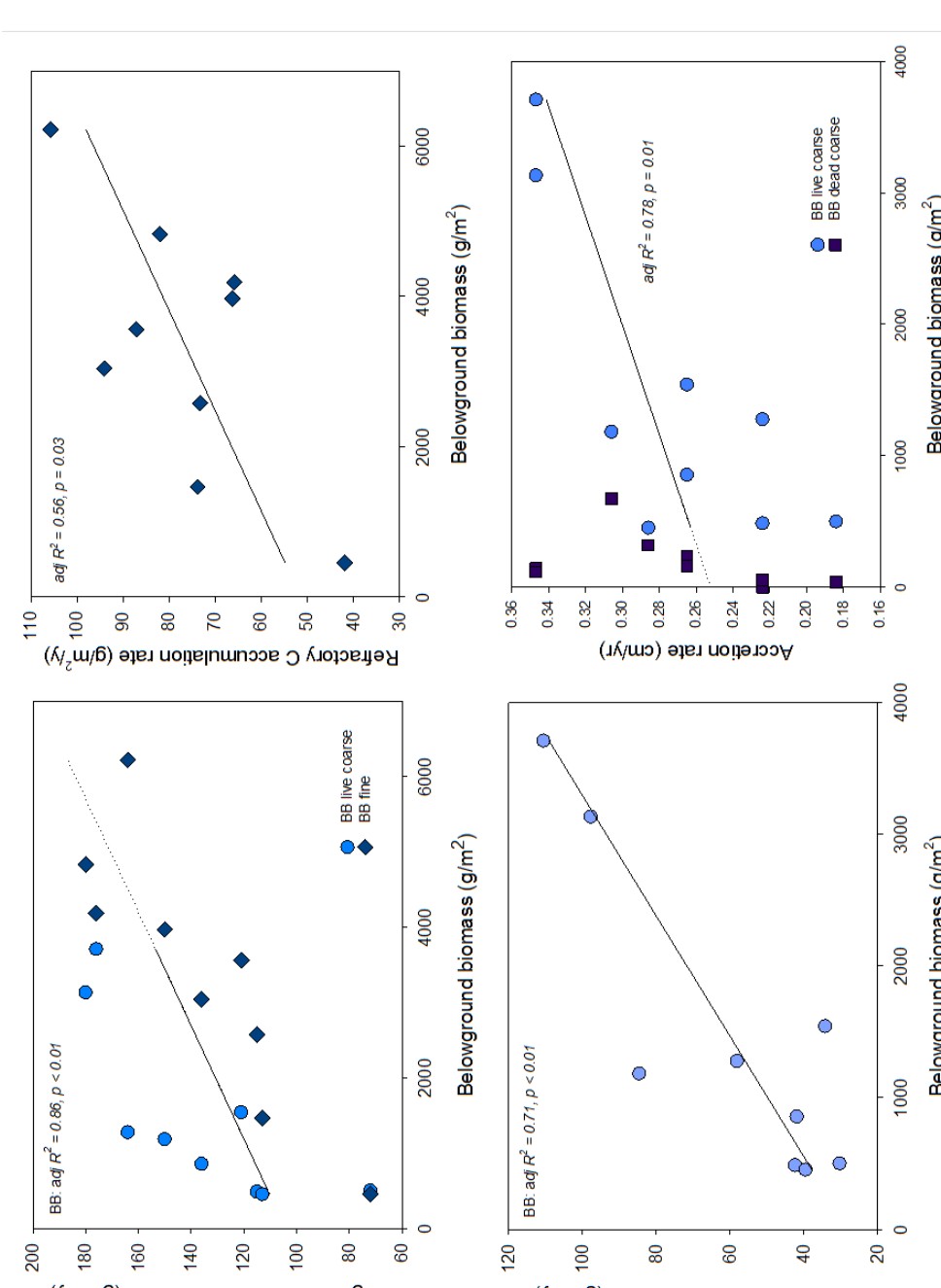

**Figure 9:** Relationship between belowground biomass structures and organic C (total, labile and refractory) accumulation and accretion rate in marshes of Barnegat Bay (BB) and Delaware Bay (DB). Only significant (p < 0.05) relationships are shown.





**Supplemental Table A. Regression model results for the relationship between hydrology variables and surface accretion and accumulation rates in marshes of two mid-Atlantic US estuaries.**

| Response variable | Predictor variable | Adjusted R-square | t Ratio | P > F |
|---|---|---|---|---|
| Surface accretion rate | MHW | 0.50 | 3.20 | 0.0064 |
| | MLW | | -4.26 | 0.0008 |
| Organic accumulation rate | MHW | 0.50 | 3.30 | 0.0054 |
| | MLW | | -3.69 | 0.0024 |
| Mineral sediment accumulation rate | MHW | 0.43 | 2.86 | 0.0126 |
| | MLW | | -3.75 | 0.0022 |