# Peer review of "Geomorphic influences on the contribution of vegetation to soil C accumulation and accretion in *Spartina alterniflora* marshes"

_Biogeosciences, 2017_

## Referee Comment (RC1) · Anonymous Referee #1 · 11 Sep 2017

This manuscript covers a lot of topics and presents some interesting data, but it is difficult to follow and could be more effective in highlighting the key findings.

Three hypotheses are presented but are poorly linked to data collection and analysis (they seem like after the fact general points rather than truly testable hypotheses). From the manuscript, the hypotheses are: (1) environmental parameters are highly correlated across marshes; however, hydrology is the most important predictor of belowground productivity, decay rates, and above- and belowground biomass; (2) short-term (< 2 yr) surface accretion rates are influenced by a combination of aboveground vegetation structure, belowground productivity, decay and mineral sedimentation rate;

and (3) longer-term (∼50 years) accretion and soil C accumulation are more strongly related to belowground biomass in organogenic marshes in a coastal lagoon than in more minerogenic marshes of a coastal plain estuary, where the potential for alloc-thonous C contributions are greater.

When I first read these, I wondered how would these be tested? For #1: how can you determine that hydrology is the most important predictor (in part, there are many components of "hydrology", how do you determine relatively importance, and in com-parison to what other factors?). For #2: this seems very open ended rather than a testable hypothesis: accretion rates are influenced by a combination of factors? And for #3, the authors come to this conclusion in the discussion, but no mention is made in the analyses of how these comparisons would be made (what is the data/statistical support for this). The manuscript would be much more effective and focused, if clear, testable hypotheses were presented. The data collection and analyses should clearly identify how these hypotheses are to be tested. This would give some structure to the results rather than the wide ranging review of results that currently are difficult to link to specific questions/hypotheses.

There are some arguments that are presented that are difficult to untangle: for exam-ple, mineral matter drives productivity. If this is the case, what is the expected "re-sponse" that supports this and what "response" would not support this (how would the measured parameters of total biomass, ingrowth, mineral matter accumulation, etc., vary if this is true and what if it is not true – or is the key in the the relationships of different parameters)? As above a clear articulation of expectations (hypotheses) is essential but lacking. Without these, it's an interesting story but not so clear what is actually being supported from these findings/data.

Part of my confusion in interpreting the results is that this is a relatively complex set of experiments with many different factors and response parameters. In terms of factors, there are two locations, with multiple sites within each location – and many factors vary both across locations as well as within locations: tidal range, sediment inputs, salinity,

inundation, etc. . . Plus there are many different response parameters, some closely related, some not (ingrowth, biomass, decomposition, accretion, C accumulation, mineral accumulation, etc. . .). It might be very useful to put together a summary table that links the various components of this research to the hypotheses/research questions of interest (factors, locations/sites, responses, expectations). Or at a minimum, to clearly identify in the methods what these links are: to test the hypothesis #2, we compared xxxx across sites (or across inundation conditions within sites), using xxxx analysis. . .

Overall, I found the writing difficult to follow. Many of the paragraphs are very long and cover a mix of topics. I'd suggest focused paragraphs with very clear topic sentences so that the logic of each section is clear and easier to follow than the current paragraphs that ramble over a mix of topics. In addition, there are some grammatical mistakes, dropped words, etc. that make the manuscript difficult to understand (e.g., l.66: should be wide range OF geomorphic settings)—check throughout for grammar (many compound sentences missing commas (l.212-213), etc. . .). Also some sentences are overly complex and difficult to follow (for example, the last sentence of the abstract): "These findings indicate that mineral sedimentation is of utmost importance for promoting belowground biomass and soil C accumulation in sediment-limited systems while in minerogenic systems, belowground biomass may not scale with C accumulation and accretion, which may be influenced more by smaller submillimetre-sized C particles." (secondarily, I don't think submillimeter particles are brought up again in the manuscript, so why are they in the abstract?)

It was not entirely clear what was previously collected background information, and what was new data for this study. For example, you refer to published rates of accretion from Boyd et al. 2017: are the accretion rates here the same data or different?

The discussion of elevation is not so clear. Be more specific. I'm assuming that it is relative elevation that is critical (where within the tidal frame the marsh surface is found). For example on Figure 4, is this elevation relative MLW (see other point below about MLW)? And are positive elevations above or below MLW? I would put lower

elevations on the left side of the x-axis (not sure if this is the case as presented). It also looks like much of this relationship is driven by the two points with zero biomass. How does this affect your interpretation: is it just a threshold relationship or is it really a linear relationship?

Also, for figure 5, organic matter inventory: the one outlier seems to be driving this relationship. Does this affect your interpretation?

It's surprising that you've found a strong fit between mineral accumulation and accretion rather than organic matter and accretion. Most others have found differently (e.g., Turner et al. 2000). How can you explain this difference?

For Figure 3: how can the decay rates and the % mass remaining not be indirectly related: How can CC have the highest decay rate, but have more mass remaining than 3 of the other sites? These should be strongly related.

Many of the figures present multiple panels, and it is not clear, what is essential to get out of a figure: seems more like a fishing expedition in presenting a wide range of results rather than targeting specific questions/hypotheses.

Details: You refer to cores of 6 cm diam. in line 144, but then 15 cm cores in l. 160. Were two different sets of cores taken? This needs to be clarified.

Be consistent in how you refer to sites: sometimes in the coastal plain site, sometime it's a minerogenic site.

paragraph starting at l.276 (and paragraph above): this all seems very exploratory, with little focus: you looked at a wide range of variables for patterns, went with MHW and MLW. As above, link the approach to the hypotheses (and move the methods to the methods section and out of the results). Also, it was not clear to me how MHW and MLW represent the range of factors (were these absolute elevations of MHW or MLW) – as you can see, I did not follow this section of the ms. very well (it was not clear to me, but maybe it is clear to others). Similarly at l.288: how does "MLW influence

ingrowth rates"? MLW is a characteristic of a particular site, but how does it influence growth across a marsh?

Lead with the key issues in presenting the data for each section. For example, for aboveground vegetation structure (paragraph starting at l. 316): clearly stem density is important, but why include the CV here: what is the significance of this? As above, I got lost in the details of the data that were presented, and did not see the key issues from the results.

Figure 1: provide some context. Not all readers know where Delaware and New Jersey are.

Other figures: As above, be consistent in mentioning features of sites so people will remember lagoon vs. coastal, minerogenic. . ... For example for Figure 3, group sites as you do for Figure 2 (or color bars or use hatching so that the two groups are obvious).

Figure 3: the dark bars on the bottom panel, make it very difficult to see the symbols for organic matter accumulation rates.

Similar to the broader point about figures with multiple panels, some multi-panel figures are not organized intuitively (at least not for this reader). For example, for Figure 7: what is mineral sedimentation the x-axis on the top two LEFT panels and the bottom two RIGHT panels? The wide mix of combinations, makes if very difficult to see patterns and follow the logic of the data presentation.

References: Some of them are out of order: See Cahoon at l. 618 and again at l. 639. In addition, some journal titles are abbreviated, some are spelled out in full (l.619 & 625).

---

## Referee Comment (RC2) · Anonymous Referee #2 · 20 Sep 2017

The authors conducted a study to investigate the impact of environmental conditions across marshes on biomass, belowground production, sediment accretion, organic/mineral accumulation. The scientific questions addressed by the ms fall within the scope of BG. The authors examined different belowground processes, and related them to each other and biogeochemical processes. The study will present some interesting results for the studies of saltmarsh sediment accretion and carbon sequestration after careful revision. General comments This study used many data from paralleled studies, such as Unger et al., (2016) and Boyd et al., 2017. To avoid confusion, you need to clearly show which data come from paralleled studies. Data analyses need to be checked and refined. Tidal range and mean water level are calculated from mean

low water and mean high water, organic/mineral accumulation rate is calculated from sediment accretion rate. You can not do correlation or regression analysis between the variables and those variables they are calculated from. The significant difference should be labelled alphabet-sequentially. Specific comments Abstract Line 7: add of after rates. Line 14: add permil after 7-40. 1. Introduction Line 29-31: you need to add references to support your statement, such as Ouyang et al. (2017). Line 59-61: I suggest you add some references here, such as Haslett et al. (2003). Line 66: add of after range Line 99-100: The allochthonous source of labile C may also include C input from riverine sources where marshes are near rivers or delta. see Craft (2007) 2. Methods Line 162: remove the after each. Line 184: what's the diameter of coarse roots and rhizomes used in your study? Line 196-204: from your results, I understand you quantified belowground biomass to both 50cm depth and the maximum of Cs-137 profile. Please clarify this point clearly here. Line 214: Some mineral material may be lost during from high temperatures of LOI analysis. Have you done acid treatment to remove inorganic carbon before LOI analysis? Line 207: Please specify the month of start and end periods. Line 208-10: The justification of longer periods for accretion estimation may also lie in the fact that organic matter accretion lags behind belowground ingrowth as it takes some time for the newly grown roots to decompose. Line 225-7: Have you conducted the homogeneity test before ANOVA or MANOVA? 3 Results Table 1: add the statistical method you used in comparison of the variables. Please check the label of 'Mean high water'; you have ab, bc, d, cd, a, ab but it is weird that there is no c. Some other variables also have the same problem, such as 'tidal range' and 'long-term mineral sedimentation rate' . Normally, the labels should be a, ab, b, bc, c...... You also need to check flooding events and duration of floods. For example, IB has 24 flooding events but 324h (per month or year?) of flooding time while MR has 455 flooding events but 7h of flooding time. Table 2: I suggest you remove tidal range and MWL in the correlation analysis, or you keep them and remove MHW and MLW, and modify your results in '3.1'. Tidal range is the difference between MHW and MLW, while MWL is the mean of MHW and MLW. You cannot correlate MHW or MLW with tidal range

and MWL just like you will not correlate the area of a circle with the diameter (A=d2/4) since this is common sense. Figure 3: why you do not show organic matter accumulation rate for IB? Line 294-6: you analyzed decay constant (Figure 3 and 4) rather than decay rates, and need to keep consistency in context. Line 309: were related or were not related? The sentence means they are related since you used 'neither' and 'nor'. Line 298-9: the last sentence is unnecessary if these variables are excluded in the stepwise regression analysis. Figure 5 caption: the dependent variables in your regression analysis should not be organic/mineral accumulation rates, of which the unit is g/(m2.yr). The accumulation rates in Table 1 are the correct term. You need to revise '3.2.3' accordingly. Table 4: why don't you use the data from all the sites to conduct the analysis of labile/refractory C density∼belowground biomass? Figure 7: significant outliers are found in the relationship live belowground coarse biomass∼MHW. Why do you say mineral sedimentation rates correspond to average rates over the last 50 years? The time dated using Cs-137 relates to nuclear events (e.g.1963). Since sediment accretion rates vary from site to site and even position to position within the same site, the dating time at 50cm depth may not all be 50 years. Figure 8: a typo in the caption. It should be Table 3 instead of Table 2. Figure 9: No relationships between belowground biomass and (organic, refractory, labile) C accumulation/accretion are shown for Delaware Bay. Are all the relationships insignificant? Have you considered to examine the relationship between C accumulation rate (as a whole, rather than organic, refractory, labile) and belowground biomass? Discussion Line 402: you need to be specific about decay. It is decay constant. Line 403: add mineral before sedimentation rate. Line 404: add coarse before belowground biomass. Line 405: replace little with insignificant as you can not consider the insignificant relationship in the linear regression analysis as little influence. Maybe environmental parameters co-vary with other factors, and explain some variance in multiple regression analyses. Line 420: I only found you examined the relationship between biomass and MHW. Where do you analyze the influence of elevation on biomass? Line 432-3: tidal range is not comparable based on your results. One is labelled bc and the other is ab. Line 446: the

explained variance is 58% rather than 62%. Line 454: it is decay constants rather than decay rates which you did not estimate in your results. You need to modify other parts of the ms accordingly. Line 484-7: There' are no direct linkage between CO2 emissions and decay rates although decay contribute to CO2 emissions, since other sources also contribute to CO2 emissions such as crab burrows. Line 538-9: the factors relate to surface accretion are organic matter inventory and mineral sediment inventory. Line 551: Some sites have fine biomass lower than MR site such as RC. Line 553-4: you only show the influence of belowground biomass on specific components of C accumulation rates (organic, refractory, labile), and your discussion here and hereafter should be more specific.

References Boyd, B., Sommerfield, C.K., and Elsey-Quirk, T.: Hydrogeomorphic influences on salt marsh sediment accumulation 610 and accretion in two estuaries of the U.S. Mid-Atlantic coast. Mar. Geol., 383, 132-145, 2017. Unger, V., Elsey-Quirk, T., Sommerfield, C. and Velinsky, D. J.: Stability of organic C accumulating in Spartina 805 alterniflora-dominated marshes of the mid-Atlantic U.S.A. Estuar. Coastal Shelf Sci. 182: 179-189, 2016. Ouyang, X., Lee, S. Y., Connolly, R.M. (2017) The role of root decomposition in global mangrove and saltmarsh carbon budgets. Earth-Science Reviews.166: 53-63. Haslett, S. K., Cundy, A. B., Davies, C. F. C., Powell, E. S., & Croudace, I. W. (2003). Salt marsh sedimentation over the past c. 120 years along the west Cotentin coast of Normandy (France): relationship to sea-level rise and sediment supply. Journal of coastal research, 609-620. Craft, Christopher. "Freshwater input structures soil properties, vertical accretion, and nutrient accumulation of Georgia and US tidal marshes." Limnology and oceanography 52.3 (2007): 1220-1230.

---

## Referee Comment (RC3) · Anonymous Referee #3 · 6 Oct 2017

This paper is a substantial and interesting addition to the literature and I think that it could be publishable for Biogeosciences Discussions pending some revisions. The study reports correlations between geomorphic variation in variables such as tidal elevation, tidal frame, suspended sediment, salinity, etc. with important biotic variables affecting carbon sequestration (above and below ground biomass, root addition, root-shoot ratio, and recalcitrant/labile carbon fraction), and observations of Carbon Accumulation Rate, measured by radiocesium date and previously published by Unger et al. and Boyd et al.. The observations that complex correlations between root production and drainage, and sediment trapping having multiple positive interactions with carbon burial, are compelling conclusions as they point to the complex and dynamic nature of

tidal wetland systems more generally. Predicting behavior outside of well-studied plots and sites is a large concern of the community and this paper speaks to many difficulties in those efforts.

I think what set's this paper apart from much of the literature is how well monitored all of the sites are. All locations have measured elevation, inundation, and soil properties. This should be commended and in many ways is close to an ideal salt marsh carbon dynamics study design.

I have thee major critiques of the paper, somewhat overlapping. 1. There is not enough available methods data for the calculation of tidal datums from the water loggers. I found some of the inclusion of comparing NAVD88 elevation, MHW and MLW hard to follow, especially when these were used as proxies for multiple hydrologic properties. I was a bit taken aback at how much variation there were in datums that are located fairly close together. Could this be because of the short 1.5-year time period? 'No data' values deflating MLW datums? Etc? Is there really that much local variation in datums? I would like to see more information before making a judgment there.

2. This leads me to my second critique. A lot of the correlation analysis could be paired down. First because of what I discussed in one, maybe some of these measures are redundant or could be reduced to more directly causal variables.

For example, by converting site elevations (NAVD88) into dimensionless elevation $z = $ (Elevation - MTL) / (MHHW-MTL) [Swanson et al., 2012], or focusing on flooding depth and # of floods a year (maybe converted to volume of water / unit time) since those are probably a clearer functional driver for organic and inorganic sedimentation. Maybe there's a better metric for drainage, such an average time between inundation events.

The number of correlations discussed makes the text — especially figure 7 — a bit cluttered and hard to decipher on a quick read through (although there are benefits to being thorough). Statistically, the number of correlations presented is problematic. Which leads me to point 3.
3. There are many correlations presented, but the threshold for significance (0.05) is not adjusted for multiple testing. The more correlations your run, the higher chance of getting false positives. Researchers often address this by using the Bonferroni correction, or some related correction (there are some arguments that the Bonferroni is too restrictive). The fact that many of the regressions presented are barely significant at the 0.05 level and many would no longer be significant after correcting for multiple testing. Maybe a combination of reducing the number of variables tested to a few functionally important variables and adjusting for multiple testing could allow the authors to focus more on the very clearly significant driving geomorphic variables.

Line comments: 63: This is an excellent point that does not get enough attention in the literature. 64: Is there any literature you could cite in the hazards literature or other ecosystem-climate change dynamics that discuss complexities in projecting system resilience? 132: What is the time frame of RSLR? Same as cores, or total length of the gauging period? 277: If dimensionless elevation may be a better fit than using MHW and MLW. Alternatively inundation time, the number of inundation events or cumulative annual mass of water seem like they would be much better variables to use as there is process-knowledge involved. 427: Is sediment trapping by biomass part of this positive feedback?

Tab 1 - What is MHW MWL and MLW relative to NAVD88? Station datum? Tab 3 - Far left column a bit hard to read. Maybe fix in formatting. Fig. 1 - Howe a basemap with better definition. Maybe one that emphasizes the differences between uplands systems and wetlands. I would delete the service layer credits and put it into the figure caption. The map looks low resolution.

Citations: Swanson, Kathleen M., et al. "Wetland accretion rate model of ecosystem resilience (WARMER) and its application to habitat sustainability for endangered species in the San Francisco Estuary." Estuaries and Coasts 37.2 (2014): 476-492.

Abdi, H. (2007). Bonferroni and Šidák corrections for multiple comparisons. Encyclopedia of measurement and statistics, 3, 103-107.

---

## Author Comment (AC1) · 18 Oct 2017

Dear Editor,

We very much appreciate the constructive comments and suggestions from the three reviewers. We have revised the manuscript to address all of the reviewer comments, which has resulted in a much improved paper overall. Below please find the Reviewer Comments and our responses below each comment . We have also added the revised manuscript as a Supplementary PDF.

Best regards,

Tracy

REVIEWER # 1 This manuscript covers a lot of topics and presents some interesting data, but it is difficult to follow and could be more effective in highlighting the key findings. Three hypotheses are presented but are poorly linked to data collection and analysis (they seem like after the fact general points rather than truly testable hypotheses). From the manuscript, the hypotheses are: (1) environmental parameters are highly correlated across marshes; however, hydrology is the most important predictor of belowground productivity, decay rates, and above- and belowground biomass; (2) short-term (< 2 yr) surface accretion rates are influenced by a combination of aboveground vegetation structure, belowground productivity, decay and mineral sedimentation rate; and (3) longer-term (âĹij50 years) accretion and soil C accumulation are more strongly related to belowground biomass in organogenic marshes in a coastal lagoon than in more minerogenic marshes of a coastal plain estuary, where the potential for allocthonous C contributions are greater. When I first read these, I wondered how would these be tested? For #1: how can you determine that hydrology is the most important predictor (in part, there are many components of "hydrology", how do you determine relatively importance, and in comparison to what other factors?). For #2: this seems very open ended rather than a testable hypothesis: accretion rates are influenced by a combination of factors? And for #3, the authors come to this conclusion in the discussion, but no mention is made in the analyses of how these comparisons would be made (what is the data/statistical support for this). The manuscript would be much more effective and focused, if clear, testable hypotheses were presented. The data collection and analyses should clearly identify how these hypotheses are to be tested. This would give some structure to the results rather than the wide ranging review of results that currently are difficult to link to specific questions/hypotheses.

We revised the text of the manuscript to improve the overall readability and provide a clear focus for the study. The reviewer is correct, the hypotheses included in the original manuscript were post-hoc and, therefore, poorly constructed. We refocused our manuscript around our a-priori (pre-study) hypotheses, which are directly related to the data analyses and results. The revised hypotheses are outlined in L 102 – 119 of the revised manuscript. Specifically, we hypothesized that rates of S. alterniflora belowground productivity were greater in marshes of the coastal plain estuary than in the marshes of the coastal lagoon, where a higher water table, higher salinity, and lower rates of sediment deposition were predicted to limit root and rhizome growth. We predicted that patterns of belowground productivity and turnover would mirror those of longer-term total and labile organic carbon accumulation rates across marshes and estuaries. Our hypothesis would be supported if environmental conditions that promoted C accumulation such as high rates of mineral sedimentation and, potentially, high tidal range and low salinity also promote high belowground biomass production. Further, we examined the role of belowground decay in explaining spatial patterns of C accumulation. We hypothesized that the amount of organic material remaining following 20 months of belowground decomposition would be greater in marshes with higher C accumulation rates. For this, the conditions that promote high rates of C accumulation may also promote the preservation of C particularly in the upper soil column where much of the decay of labile organic matter occurs (Hackney and de la Cruz 1980; Hackney 1987; Morris and Bowden 1986). Ultimately, the net amount of belowground biomass (C fractions greater than $\sim$1 mm in size) was predicted to be directly and positively related to the density of C in the soil profile and C accumulation rate. Similarly, above-and belowground biomass was predicted to be positively related to soil C accumulation. Finally, because plant productivity and decay processes as well as overall plant structure (e.g., height, stem density, biomass) have been shown to be tightly regulated by abiotic factors, we examined the influence of local environmental conditions (i.e., water level, salinity, soil nutrient status, and sediment deposition rates) on S. alterniflora growth, decay and biomass across marshes and estuaries.

There are some arguments that are presented that are difficult to untangle: for example, mineral matter drives productivity. If this is the case, what is the expected "response" that supports this and what "response" would not support this (how would the measured parameters of total biomass, ingrowth, mineral matter accumulation, etc., vary if this is true and what if it is not true – or is the key in the the the relationships of different parameters)? As above a clear articulation of expectations (hypotheses) is essential but lacking. Without these, it's an interesting story but not so clear what is actually being supported from these findings/data. Part of my confusion in interpreting the results is that this is a relatively complex set of experiments with many different factors and response parameters. In terms of factors, there are two locations, with multiple sites within each location – and many factors vary both across locations as well as within locations: tidal range, sediment inputs, salinity, inundation, etc

By revising the hypotheses, the somewhat complex study design and findings are clarified. We have also revised much of the text to provide justification for our hypotheses and context for our findings. The specific argument highlighted above, "mineral matter drives productivity" was removed in the revision of the manuscript.

Plus there are many different response parameters, some closely related, some not (ingrowth, biomass, decomposition, accretion, C accumulation, mineral accumulation, etc ...). It might be very useful to put together a summary table that links the various components of this research to the hypotheses/research questions of interest (factors, locations/sites, responses, expectations). Or at a minimum, to clearly identify in the methods what these links are: to test the hypothesis #2, we compared xxxx across sites (or across inundation conditions within sites), using xxxx analysis...

We revised the text throughout the manuscript to simplify the hypotheses and how they relate to the response variables. For example, the multivariate correlation analyses were removed from the results section and relevant information was summarized in the data analysis section. Table 2 showing the correlation analysis results was moved to a Supplementary Table.

Overall, I found the writing difficult to follow. Many of the paragraphs are very long and cover a mix of topics. I'd suggest focused paragraphs with very clear topic sentences so that the logic of each section is clear and easier to follow than the current paragraphs that ramble over a mix of topics. In addition, there are some grammatical mistakes, dropped words, etc. that make the manuscript difficult to understand (e.g.,l.66: should be wide range OF geomorphic settings) check throughout for grammar (many compound sentences missing commas (l.212-213), etc ...). Also some sentences are overly complex and difficult to follow (for example, the last sentence of the abstract): "These findings indicate that mineral sedimentation is of utmost importance for promoting belowground biomass and soil C accumulation in sediment-limited systems while in minerogenic systems, belowground biomass may not scale with C accumulation and accretion, which may be influenced more by smaller submillimetre-sized C particles." (secondarily, I don't think submillimeter particles are brought up again in the manuscript, so why are they in the abstract?)

The text was revised throughout to improve readability and grammar. Strong topic sentences were added to the beginning of paragraphs and the amount of rambling was minimized. The sentences highlighted by the reviewer above, were ultimately removed from the revised manuscript.

It was not entirely clear what was previously collected background information, and what was new data for this study. For example, you refer to published rates of accretion from Boyd et al. 2017: are the accretion rates here the same data or different?

We added text to both the Introduction (L 78 – L 99) and Methods (L 155 – L 157) to clarify that we used C accumulation rates published in Unger et al. 2016 and Cs-137 based accretion rates published in Boyd et al. 2017 for our examination of how vegetation dynamics relate to soil C dynamics.

The discussion of elevation is not so clear. Be more specific. I'm assuming that it is relative elevation that is critical (where within the tidal frame the marsh surface is found). For example on Figure 4, is this elevation relative MLW (see other point below about MLW)? And are positive elevations above or below MLW? I would put lower elevations on the left side of the x-axis (not sure if this is the case as presented). It also looks like much of this relationship is driven by the two points with zero biomass. How does this affect your interpretation: is it just a threshold relationship or is it really a linear relationship? Also, for figure 5, organic matter inventory: the one outlier seems to be driving this relationship. Does this affect your interpretation?

We have clarified our use of elevation and water level data throughout the methods and results. While we collected elevation data (relative to North American Vertical Datum 1988), what we used in our analyses were water level data, which were calculated relative to the marsh surface. This, as the reviewer indicated, is what is critical for driving plant and soil processes. For Figure 4 (Figure 3 in revised manuscript), we revised the axis to read MLW depth relative to marsh surface (cm). While it appears that the relationship was driven by two points, when those points were removed, the linear relationship remained significant with only a slight reduction in the R-square value. We included this information in the Results section, as well as a sentence on this relationship really being more of a threshold relationship, as the reviewer aptly pointed out.

It's surprising that you've found a strong fit between mineral accumulation and accretion rather than organic matter and accretion. Most others have found differently (e.g., Turner et al. 2000). How can you explain this difference?

This is a good point, which we subsequently worked into the discussion. Actually, Table 2 in Turner et al. 2000 illustrates regional differences in the contribution of mineral sediment to accretion. Along the U.S. Atlantic coastal plain, accretion rates were directly related to both mineral sedimentation and organic matter accumulation rates for most marshes (all but 1 study), as well as across all Atlantic coastal marshes combined. Our previous studies have shown a similar trend (Unger et al. 2016; Boyd et al. 2017). Conversely, in U.S. Gulf coast marshes, accretion rates were related to organic matter accumulation rate only. Turner et al. 2000, hypothesized that at high rates of mineral sedimentation, the relationship between accretion and mineral sedimentation becomes variable associated with a threshold of organic production at high rates of mineral sedimentation. I might suggest that these linear averaged accumulation rates, don't account for major declines in mineral sediment over time, and thus with a high historic sediment input and a much lower recent sediment supply to many Gulf Coast marshes, the relationship between accretion and sediment input becomes variable. In addition, accretion rates also respond to changes in relative sea-level and have done so in Gulf Coast marshes mostly by organic matter accumulation. These marshes are experiencing subsidence and deterioration due to the lack of sufficient mineral sediments to support plant growth and biomass. Our study illustrates this point, and suggests that allochthonous C burial and C preservation may also be significant at high rates of sedimentation, provided that marshes have a relatively continuous supply of sediment over time.

For Figure 3: how can the decay rates and the % mass remaining not be indirectly related: How can CC have the highest decay rate, but have more mass remaining than 3 of the other sites? These should be strongly related.

We have included a Supplementary Figure (A) to explain this. Marshes in both estuaries had similar amounts of organic matter remaining, yet Barnegat Bay had a steeper decline in organic matter over time. However, the reviewer's comment highlighted the need for some additional explanation. Litterbags in Barnegat Bay were placed in the marsh slightly later in the year than those in Delaware Bay due to logistical delays following Hurricane Sandy. As a result, asymptotic decay rates were greater in Barnegat Bay, however, this was likely associated with slightly warmer temperatures following deployment as compared to Delaware Bay. We have added this information to the Methods (L 203 – L 206) and removed the decay rate calculation and results, and instead just used % mass remaining following 20 months as the response variable.

Many of the figures present multiple panels, and it is not clear, what is essential to get out of a figure: seems more like a fishing expedition in presenting a wide range of results rather than targeting specific questions/hypotheses.

Each figure in the revised version of the manuscript is directly tied to hypotheses stated in the Introduction, data analyses in the Methods section, and findings in the Results section.

Details: You refer to cores of 6 cm diam. in line 144, but then 15 cm cores in l. 160. Were two different sets of cores taken? This needs to be clarified.

The cores were 15.5 cm in diameter. The size was corrected in the revision.

Be consistent in how you refer to sites: sometimes in the coastal plain site, sometime it's a minerogenic site.

We revised the manuscript for consistency when referring to estuaries and marshes. We refer to coastal plain and coastal lagoon in the Introduction and Discussion. We refer to the specific estuaries (i.e., Delaware Bay and Barnegat Bay) in the Methods and Results sections.

Paragraph starting at l.276 (and paragraph above): this all seems very exploratory, with little focus: you looked at a wide range of variables for patterns, went with MHW and MLW. As above, link the approach to the hypotheses (and move the methods to the methods section and out of the results).

This section was removed in the revision. The correlation analysis was performed so as to select environmental variables that were somewhat independent, not strongly co-varying with others, so as to limit our interpretation. We moved any relevant information to the Data Analysis section.

Also, it was not clear to me how MHW and MLW represent the range of factors (were these absolute elevations of MHW or MLW) – as you can see, I did not follow this section of the ms. very well (it was not clear to me, but maybe it is clear to others). Similarly at l.288: how does "MLW influence ingrowth rates"? MLW is a characteristic of a particular site, but how does it influence growth across a marsh?

This point is now clarified in the revised manuscript. MHW and MLW were average high and low water depths relative to the marsh surface calculated from two years of continuous water level data. Root ingrowth rates were higher in marsh locations where the average low water depth was lower than in areas where the average low water depth was high (i.e., greater root growth with greater drainage during low tide).

Lead with the key issues in presenting the data for each section. For example, for aboveground vegetation structure (paragraph starting at l. 316): clearly stem density is important, but why include the CV here: what is the significance of this? As above, I got lost in the details of the data that were presented, and did not see the key issues from the results.

We highlighted key issues and findings relevant to the hypotheses to the beginning of each result section.

Figure 1: provide some context. Not all readers know where Delaware and New Jersey are.

We added context to the legend.

Other figures: As above, be consistent in mentioning features of sites so people will remember lagoon vs. coastal, minerogenic.... For example for Figure 3, group sites as you do for Figure 2 (or color bars or use hatching so that the two groups are obvious).

As the reviewer suggested, we have revised figures to be consistent using Barnegat Bay and Delaware Bay for designation of data points.

Figure 3: the dark bars on the bottom panel, make it very difficult to see the symbols for organic matter accumulation rates.

This figure was deleted in the revision of the paper.

Similar to the broader point about figures with multiple panels, some multi-panel figures are not organized intuitively (at least not for this reader). For example, for Figure 7: what is mineral sedimentation the x-axis on the top two LEFT panels and the bottom two RIGHT panels? The wide mix of combinations, makes if very difficult to see patterns and follow the logic of the data presentation.

We revised Figure 7 (Figure 4 in revised manuscript) such that all x-axis mineral sedimentation rates are on the left panel and all x-axis MHW depth are on the left panel, as the reviewer suggested.

References: Some of them are out of order: See Cahoon at l. 618 and again at l. 639. In addition, some journal titles are abbreviated, some are spelled out in full (l.619 &625).

We have edited the references, as suggested by the reviewer.

Please also note the supplement to this comment:
https://www.biogeosciences-discuss.net/bg-2017-268/bg-2017-268-AC1-supplement.pdf

—————————————————————

[revised manuscript text omitted]
 influenced C accumulation through its positive relationship with labile C accumulation rate.
Aboveground biomass was stimulated in the coastal plain estuary by greater flooding, ultimately leading to greater
labile and total C accumulation rates.  As discussed below, the positive relationship between aboveground biomass
and labile C accumulation rate may result from labile tissue production as a function of photosynthetic area and/or the
increased trapping and deposition of allochthonous labile C such as algal-derived C.  Relatively high densities of labile
C in the mineral-rich marshes, independent of belowground biomass, indicate potential allochthonous contributions
and high preservation of labile C.

[revised manuscript text omitted]

Rates of soil C accumulation and accretion were strongly related to *S. alterniflora* biomass. Aboveground live biomass was positively related to organic C accumulation rate across estuaries, primarily though the strong relationship with labile C accumulation rate. Despite the potential for high inter-annual variability in peak biomass aboveground (Gross et al., 1990), spatial variation in peak aboveground biomass in marshes of the same region may be conserved over
time. Mechanisms of aboveground live biomass influencing labile soil C include the direct input of aboveground litter
to the soil. For *S. alterniflora* much of the aboveground litter is subject to decay and mechanical breakdown by tidal
action (Teal 1962), and therefore, little of the aboveground litter of *S. alterniflora* in these temperate marshes is thought
to be incorporated into the marsh soil. The standing live aboveground biomass, however, represents both
photosynthetic capacity and growing conditions, which may directly affect the production of labile exudates and new
labile tissues belowground. Additionally, aboveground shoots influence surface deposition and accumulation through
the direct capture of particles on plant stems and the indirect reduction of flow velocity inducing sediment settling
(Stumpf 1983; Leonard and Luther 1995). This has been illustrated for fertilized and unfertilized *S. alterniflora* plots,
where aboveground biomass was three times greater and surface accretion rate was 2 mm/yr greater in response to
fertilization (Morris and Bradley 1999; Morris et al. 2002). The higher accretion rate was accounted for by more
efficient trapping of sediments (Mudd et al., 2010). Therefore, both labile live plant tissues and labile organic C
associated with sediment may be enhanced with greater aboveground biomass. This finding supports other studies
showing positive relationships between aboveground biomass and soil microbial processes, including the
decomposition of recalcitrant soil organic matter, which was hypothesized to be due to greater labile C inputs
(substrate-induced priming) and/or greater rhizosphere oxygenation ($O_2$-induced priming; Mueller et al. 2016). Our
data suggest that aboveground biomass may increase soil C accumulation directly through the inputs of labile C and
positive bio-physical feedbacks for sedimentation, which increases labile C deposition, burial, and preservation.
Relationships among aboveground plant biomass, labile and total C accumulation rate, and mineral sedimentation rate
indicate positive feedbacks among these processes (Unger et al. 2016; present study).

Aboveground biomass response to flooding was estuary-dependent. In the coastal plain estuary, aboveground biomass
increased with a higher mean high tide level, whereas in the coastal lagoon, aboveground biomass declined with higher
mean high water depth. In the coastal lagoon, stem density increased with greater drainage. With all other conditions
being equal, plant biomass of *S. alterniflora* has a parabolic relationship with elevation relative to mean sea level
(Morris et al. 2002). Above- and below an optimum elevation, biomass declines. Our study of marshes in different
geomorphic settings, illustrate how environmental conditions in estuaries can illicit differential responses to individual
environmental parameters. In the coastal lagoon, a combination of less flooding and greater mineral sedimentation
rates promoted greater aboveground (and belowground) biomass. Lower tolerance to flooding in the coastal lagoon
marshes as compared to the coastal plain marshes is likely due to greater soil organic matter content, lower redox
potential, lower mineral sediment and nutrient availability, and higher sulfide concentrations (Bradley and Morris
1990; Reddy and DeLaune 2008). Because aboveground biomass was strongly associated with labile C accumulation
in marshes across the two estuaries, flooding dynamics and aboveground plant responses ultimately influenced labile
C accumulation. Limited sediment availability (e.g., RC and IB) also influenced negative feedbacks between labile

[revised manuscript text omitted]

---

## Author Comment (AC2) · 18 Oct 2017

REVIEWER #2 The authors conducted a study to investigate the impact of environmental conditions across marshes on biomass, belowground production, sediment accretion, organic/mineral accumulation. The scientific questions addressed by the ms fall within the scope of BG. The authors examined different belowground processes, and related them to each other and biogeochemical processes. The study will present some interesting results for the studies of saltmarsh sediment accretion and carbon sequestration after careful revision. General comments This study used many data from paralleled studies, such as Unger et al., (2016) and Boyd et al., 2017. To avoid

confusion, you need to clearly show which data come from paralleled studies.

As described above, we have extensively revised this manuscript. We clarified the inclusion of C accumulation rates published in Unger et al. 2016 and Cs-137 based accretion rates published in Boyd et al. 2017 for our examination of how vegetation dynamics relate to soil C dynamics. The text was added to the Introduction (L 78 – L 99) and Methods (L 155 – L 157).

Data analyses need to be checked and refined. Tidal range and mean water level are calculated from mean low water and mean high water, organic/mineral accumulation rate is calculated from sediment accretion rate. You cannot do correlation or regression analysis between the variables and those variables they are calculated from.

We conducted a correlation analysis to examine which variables (mean low water and/or mean high water) was driving variability in tidal range. While statistically, tidal range was calculated from the difference of MHW and MLW, our analysis revealed that spatial variation in tidal range was driven by differences in average high water not average low water. This illustrates how marsh interiors do not drain much at low tide and any difference in tidal range across marshes is due to high tide levels. We maintain that although this is not a main focus of the paper, it is still important.

Surface accretion and accumulation rates were removed in the revision.

The significant difference should be labelled alphabet-sequentially.

I am not exactly sure what this comment is referring to.

Specific comments

Abstract Line 7: add of after rates.

This line of the abstract was changed in the revision.

Line 14: add permil after 7-40.

Some journals show salinity is unitless; I will defer to the Editor for recommendation.

Introduction Line 29-31: you need to add references to support your statement, such as Ouyang et al. (2017).

This specific line was modified in the revised paper to: "Plant biomass, especially belowground biomass, is considered to be a primary contributor to soil organic matter and carbon (C) sequestration in marshes (DeLaune et al. 1983; Nyman et al. 2006)." References were added, as the reviewer suggested.

Line 59-61: I suggest you add some references here, such as Haslett et al. (2003).

We appreciate the reference recommendation and added it to the sentence, which is now L 49 – 51 in the revised manuscript.

Line 66: add of after range

This sentence was removed in the revision.

Line 99-100: The allochthonous source of labile C may also include C input from riverine sources where marshes are near rivers or delta. see Craft (2007)

This specific sentence was removed in the revision, however, two sentences in the revised manuscript, L 64 – 67, "Higher tidal range, greater supply of mineral nutrients and sediments, and lower salinities are conditions that are all predicted to enhance both plant productivity and soil C accumulation (Mendelssohn and Kuhn 2003; Craft 2007; Kirwan and Guntespergen 2010)." And L 72 – 74, "In contrast, marshes in geomorphic settings with high rates of mineral sedimentation such as those near river deltas may have greater magnitudes of both allocthonous C deposition and autochthonous plant C inputs (e.g., Craft et al. 2007)." include the Craft et al. 2007 citation. Again, we appreciate the reference recommendation.

2.Methods

Line 162: remove the after each.

The correction was made as the reviewer suggested.

Line 184: what's the diameter of coarse roots and rhizomes used in your study?

We did not measure the diameter of coarse roots and rhizomes.

Line 196-204: from your results, I understand you quantified belowground biomass to both 50cm depth and the maximum of Cs-137 profile. Please clarify this point clearly here.

As the reviewer suggested, we added the methods for calculating belowground biomass to both the 137Cs-peak depth and 50-cm depth.

Line 214: Some mineral material may be lost during from high temperatures of LOI analysis. Have you done acid treatment to remove inorganic carbon before LOI analysis?

These data were removed in from the revised manuscript. As the reviewers generally alluded to, there was a lot of data and analysis presented in the original manuscript, and we decided to simplify the scope and just concentrate on longer-term accretion and accumulation rates.

Line 207: Please specify the month of start and end periods. Line 208-10: The justification of longer periods for accretion estimation may also lie in the fact that organic matter accretion lags behind belowground ingrowth as it takes some time for the newly grown roots to decompose.

Yes, the reviewer points out an additional reason for removing these data and comparisons.

Line 225-7: Have you conducted the homogeneity test before ANOVA or MANOVA?

Yes, we used the Levene Test. We added this information to L 235 – 237 of the revised manuscript: " We tested for homogeneity of variances using the Levene Test on transformed data. The only violation of the equal variance assumption was for the

95% rooting depth, which, following log-transformation, failed the Levene test between estuaries, but not among marshes."

3 Results

Table 1: add the statistical method you used in comparison of the variables.

As the reviewer suggested, we added the statistical test to the legend of Table 1.

Please check the label of 'Mean high water'; you have ab, bc, d, cd, a, ab but it is weird that there is no c. Some other variables also have the same problem, such as 'tidal range' and 'long-term mineral sedimentation rate' . Normally, the labels should be a, ab, b, bc, c......

We double checked the statistical output, and the letter designations in Table 1 correctly reflect the output of the Post-hoc Tukey Test. I think there is just a lot of overlap in error between sites, which is also a function of the nested design. I have attached some output from our test of tide range differences between marshes nested in estuaries:

You also need to check flooding events and duration of floods. For example, IB has 24 flooding events but 324h (per month or year?) of flooding time while MR has 455 flooding events but 7h of flooding time.

These data are correct. IB has had 24 flooding events and the duration of each flood averaged 324 hrs. By also comparing this with the % time flooded, it is clear that IB is almost continuously flooded so the # of flooding events is low, yet the flood duration is high. MR, on the other hand, is flooded frequently but has an average flood duration of 7 hours for each flooding event.

Table 2: I suggest you remove tidal range and MWL in the correlation analysis, or you keep them and remove MHW and MLW, and modify your results in '3.1'. Tidal range is the difference between MHW and MLW, while MWL is the mean of MHW and MLW. You cannot correlate MHW or MLW with tidal range and MWL just like you will not correlate the area of a circle with the diameter $(A=d2/4)$ since this is common sense.

Results section 3.1 was revised and the correlation analysis was removed from the Results section in the revision. Please see response to the same comment above.

Figure 3: why you do not show organic matter accumulation rate for IB?

Figure 3 was removed in the revision.

Line 294-6: you analyzed decay constant (Figure 3 and 4) rather than decay rates, and need to keep consistency in context.

Decay constant and decay rates were removed in the revision.

Line 309: were related or were not related? The sentence means they are related since you used 'neither' and 'nor'.

This sentence was removed in the revision.

Line 298-9: the last sentence is unnecessary if these variables are excluded in the stepwise regression analysis.

This sentence was removed in the revision.

Figure 5 caption: the dependent variables in your regression analysis should not be organic/mineral accumulation rates, of which the unit is g/(m2.yr). The accumulation rates in Table 1 are the correct term. You need to revise '3.2.3' accordingly.

Figure 5 was removed in the revision.

Table 4: why don't you use the data from all the sites to conduct the analysis of labile/refractory C densityâĹijbelowground biomass?

As the reviewer suggested, we added an analysis using data from all of the sites, as well as site-specific data. Combined this illustrates important biomass-C density relationship across marshes, as well as, how coastal lagoon marshes have a strong refractory C-biomass relationship, but coastal plain estuary marshes do not.

Figure 7: significant outliers are found in the relationship live belowground coarse

biomassâĹ́ijMHW.

Indeed, live belowground biomass was much more strongly related to mineral sedimentation rate across marshes, which is also shown in Figure 7 (Figure 4 in revised manuscript). We added a comment about the variability in the relationship between MHW and biomass in the results section, which is reflected in the r-square value (0.44) in Figure 7 (Figure 4 in revised manuscript).

Why do you say mineral sedimentation rates correspond to average rates over the last 50 years? The time dated using Cs-137 relates to nuclear events (e.g.1963). Since sediment accretion rates vary from site to site and even position to position within the same site, the dating time at 50cm depth may not all be 50 years. This was a misstatement, and was removed in the revision.

Figure 8: a typo in the caption. It should be Table 3 instead of Table 2.

Yes, this was changed as the reviewer suggested and in the revised version, the legend refers correctly to Table 2.

Figure 9: No relationships between belowground biomass and (organic, refractory, labile) C accumulation/accretion are shown for Delaware Bay. Are all the relationships insignificant? Have you considered to examine the relationship between C accumulation rate (as a whole, rather than organic, refractory, labile) and belowground biomass?

Yes, the goal was originally to examine C-biomass relationships across estuaries. Yet, there were no significant relationships across estuaries. The only significant relationships were found in Barnegat Bay. We analyzed relationships for total organic (labile + refractory), labile, and refractory C and all biomass fractions.

Discussion

Line 402: you need to be specific about decay. It is decay constant.

This was changed in the revision, as explained above. The response variable is now

percentage of dry mass remaining.

Line 403: add mineral before sedimentation rate.

This sentence was removed in the revision.

Line 404: add coarse before belowground biomass.

This sentence was removed in the revision.

Line 405: replace little with insignificant as you can not consider the insignificant relationship in the linear regression analysis as little influence. Maybe environmental parameters co-vary with other factors, and explain some variance in multiple regression analyses.

This sentence was removed in the revision of the manuscript.

Line 420: I only found you examined the relationship between biomass and MHW. Where do you analyze the influence of elevation on biomass?

We did not analyze the relationship between biomass and elevation. We added text to the revised manuscript to explain why. In the Results section (L259 – L 266) and in the Discussion, we describe how hydrology was uncoupled to elevation across marshes due to factors such as poor drainage through mosquito ditches in IB, which created high water levels throughout the study, despite moderate elevations. Because plant and soil C processes respond more directly to hydrology than elevation relative to a datum, we used hydrologic parameters instead of elevation.

Line 432-3: tidal range is not comparable based on your results. One is labelled bc and the other is ab.

The statistical results indicate that there is not a significant difference in tidal range between RC and DN.

Line 446: the explained variance is 58% rather than 62%.

[Figure]

This sentence was changed in the revision.

Line 454: it is decay constants rather than decay rates which you did not estimate in your results. You need to modify other parts of the ms accordingly.

This was modified in the revision.

Line 484-7: There' are no direct linkage between $CO_2$ emissions and decay rates although decay contribute to $CO_2$ emissions, since other sources also contribute to $CO_2$ emissions such as crab burrows.

As the reviewer suggested, this sentence was removed in the revision.

Line 538-9: the factors relate to surface accretion are organic matter inventory and mineral sediment inventory.

This sentence was removed in the revision.

Line 551: Some sites have fine biomass lower than MR site such as RC.

This sentence was removed in the revision.

Line 553-4: you only show the influence of belowground biomass on specific components of C accumulation rates (organic, refractory, labile), and your discussion here and hereafter should be more specific.

We added some clarification throughout the manuscript that total organic C = labile C + refractory C. Thus, we examined relationships between belowground biomass and labile, refractory and total organic C accumulation rates as a whole.

References Boyd, B., Sommerfield, C.K., and Elsey-Quirk, T.: Hydrogeomorphic influences on salt marsh sediment accumulation 610 and accretion in two estuaries of the U.S. Mid-Atlantic coast. Mar. Geol., 383, 132-145, 2017. Unger, V., Elsey-Quirk, T., Sommerfield, C. and Velinsky, D. J.: Stability of organic C accumulating in Spartina 805 alterniflora-dominated marshes of the mid-Atlantic U.S.A. Estuar. Coastal Shelf

Sci. 182: 179-189, 2016. Ouyang, X., Lee, S. Y., Connolly, R.M. (2017) The role of root decomposition in global mangrove and saltmarsh carbon budgets. Earth-Science Reviews.166: 53-63. Haslett, S. K., Cundy, A. B., Davies, C. F. C., Powell, E. S., & Croudace, I. W. (2003). Salt marsh sedimentation over the past c. 120 years along the west Cotentin coast of Normandy (France): relationship to sea-level rise and sediment supply. Journal of coastal research, 609-620. Craft, Christopher. "Freshwater input structures soil properties, vertical accretion, and nutrient accumulation of Georgia and US tidal marshes." Limnology and oceanography 52.3 (2007): 1220-1230.

The format an inclusion of references was corrected in the revision.

---

## Author Comment (AC3) · 18 Oct 2017

REVIEWER #3

This paper is a substantial and interesting addition to the literature and I think that it could be publishable for Biogeosciences Discussions pending some revisions. The study reports correlations between geomorphic variation in variables such as tidal elevation, tidal frame, suspended sediment, salinity, etc. with important biotic variables affecting carbon sequestration (above and below ground biomass, root addition, root-shoot ratio, and recalcitrant/labile carbon fraction), and observations of Carbon Accumulation Rate, measured by radiocesium date and previously published by Unger et al.

and Boyd et al.. The observations that complex correlations between root production and drainage, and sediment trapping having multiple positive interactions with carbon burial, are compelling conclusions as they point to the complex and dynamic nature of tidal wetland systems more generally. Predicting behavior outside of well-studied plots and sites is a large concern of the community and this paper speaks to many difficulties in those efforts. I think what set's this paper apart from much of the literature is how well monitored all of the sites are. All locations have measured elevation, inundation, and soil properties. This should be commended and in many ways is close to an ideal salt marsh carbon dynamics study design. I have thee major critiques of the paper, somewhat overlapping. 1. There is not enough available methods data for the calculation of tidal datums from the water loggers. I found some of the inclusion of comparing NAVD88 elevation, MHW and MLW hard to follow, especially when these were used as proxies for multiple hydrologic properties. I was a bit taken aback at how much variation there were in datums that are located fairly close together. Could this be because of the short 1.5-year time period? 'No data' values deflating MLW datums? Etc? Is there really that much local variation in datums? I would like to see more information before making a judgment there.

Yes, the reviewer is correct, more explanation was needed to discuss the datums. This was similar to a comment made by Reviewer #1. We have clarified our use of elevation and water level data throughout the methods and results. While we collected elevation data (relative to North American Vertical Datum 1988), what we used in our analyses were water level data, which were calculated relative to the marsh surface. We included this information in the Methods and Results section.

2. This leads me to my second critique. A lot of the correlation analysis could be paired down. First because of what I discussed in one, maybe some of these measures are redundant or could be reduced to more directly causal variables.

For example, by converting site elevations (NAVD88) into dimensionless elevation z = (Elevation - MTL) / (MHHW-MTL) [Swanson et al., 2012], or focusing on flooding depth

and # of floods a year (maybe converted to volume of water / unit time) since those are probably a clearer functional driver for organic and inorganic sedimentation. Maybe there's a better metric for drainage, such an average time between inundation events. The number of correlations discussed makes the text âÌĘAËĞT especially figure 7 âÌĘAËĞT a bit cluttered and hard to decipher on a quick read through (although there are benefits to being thorough). Statistically, the number of correlations presented is problematic. Which leads me to point 3.

Yes, indeed. As the reviewer suggested, we revised the text and removed the multivariate correlation analyses from the results section. Relevant information was summarized in the data analysis section. Table 2 showing the correlation analysis results was moved to a Supplementary Table.

3. There are many correlations presented, but the threshold for significance (0.05) is not adjusted for multiple testing. The more correlations your run, the higher chance of getting false positives. Researchers often address this by using the Bonferroni correction, or some related correction (there are some arguments that the Bonferroni is too restrictive). The fact that many of the regressions presented are barely significant at the 0.05 level and many would no longer be significant after correcting for multiple testing. Maybe a combination of reducing the number of variables tested to a few functionally important variables and adjusting for multiple testing could allow the authors to focus more on the very clearly significant driving geomorphic variables.

The purpose of the correlation analysis was to identify environmental parameters for which other parameters co-varied. Almost all significant relationships had p-values of < 0.01. As a result, we only used a sub-set of environmental parameters in subsequent analyses. Almost all of these had correlations or regressions where the p-values were very low, and only 2 or fewer independent variables influenced response variables. Based on this, we don't feel that a Bonferonni correction is really necessary. However, we rans some exploratory analyses with Bonferonni corrected data, and found similar results to what is presented.

Line comments: 63: This is an excellent point that does not get enough attention in the literature.

Excellent! This sentence is now L57 – 59 in the revised version.

64: Is there any literature you could cite in the hazards literature or other ecosystem-climate change dynamics that discuss complexities in projecting system resilience?

While this is an excellent discussion point, it may be a little outside of the scope of this paper. Particularly, now that we have made changes to improve the clarity and focus.

132: What is the time frame of RSLR? Same as cores, or total length of the gauging period?

The time frame of RSLR was the same as the cores/ We added a few words to clarify. L 148-151 in the revision "Accretion rate in Barnegat Bay marshes (0.28 $\pm$ 0.06 cm/yr) over the last 50 – 100 years was less than the rate of relative sea-level rise over approximately the same time period (0.41 cm/yr; NOAA, Tides and Currents; in Boyd et al., 2017). In Delaware Bay, salt marsh accretion rate (0.70 $\pm$ 0.26 cm/yr) exceeded the rate of local relative sea-level rise over the same time period (0.34 cm/yr, NOAA, Tides and Currents)."

277: If dimensionless elevation may be a better fit than using MHW and MLW. Alternatively inundation time, the number of inundation events or cumulative annual mass of water seem like they would be much better variables to use as there is process-knowledge involved.

The reason why we chose to use MHW and MLW because these variables represent the magnitude of surface flooding at high tide and the magnitude of drainage at low tide. We anticipated that these would be important biologically. However, this was not very well clarified in the original submission, and therefore we added this information explicitly to the data analysis section.

427: Is sediment trapping by biomass part of this positive feedback?
Yes, an excellent point. In the revised manuscript, we highlight the importance of aboveground biomass and its relationship to labile C accumulation rate. While there are several mechanisms that can explain this relationship, sediment and allochthonous labile C trapping is one.

Tab 1 - What is MHW MWL and MLW relative to NAVD88? Station datum?

We added this information to the table and all relevant figures; it is relative to the marsh surface.

Tab 3 - Far left column a bit hard to read. Maybe fix in formatting.

We fixed the formatting of the table, as suggested by the reviewer.

Fig. 1 - Howe a basemap with better definition. Maybe one that emphasizes the differences between uplands systems and wetlands. I would delete the service layer credits and put it into the figure caption. The map looks low resolution.

We have replaced Figure 1 with a new better resolution map of the study locations.

Citations: Swanson, Kathleen M., et al. "Wetland accretion rate model of ecosystem resilience (WARMER) and its application to habitat sustainability for endangered species in the San Francisco Estuary." Estuaries and Coasts 37.2 (2014): 476-492. Abdi, H. (2007). Bonferroni and Šidák corrections for multiple comparisons. Encyclopedia of measurement and statistics, 3, 103-107.

We thank the Reviewer for the citations.

---

## Author Comment (AC4) · 31 Oct 2017

The comment was uploaded in the form of a supplement:
https://www.biogeosciences-discuss.net/bg-2017-268/bg-2017-268-AC4-supplement.pdf

---

## Author Response (AR1)

Letter to the Editor

November 27, 2017

Dear Jens-Arne Subke,

Thank you for your favorable review of our manuscript. As you have suggested, we have changed the term, "ingrowth", to root productivity or growth in the Introduction and Discussion wherever relevant. Attached please find the revised manuscript, which we are excited to have published in Biogeosciences.

Best regards,

Tracy Quirk